# Small Molecules Targeting Programmed Cell Death in Breast Cancer Cells

**DOI:** 10.3390/ijms22189722

**Published:** 2021-09-08

**Authors:** Subashani Maniam, Sandra Maniam

**Affiliations:** 1School of Science, STEM College, RMIT University, Melbourne, VIC 3001, Australia; 2Department of Human Anatomy, Faculty of Medicine and Health Sciences, Universiti Putra Malaysia, Serdang 43400, Malaysia

**Keywords:** programmed cell death, apoptosis, autophagy, breast cancer, small molecule

## Abstract

Targeted chemotherapy has become the forefront for cancer treatment in recent years. The selective and specific features allow more effective treatment with reduced side effects. Most targeted therapies, which include small molecules, act on specific molecular targets that are altered in tumour cells, mainly in cancers such as breast, lung, colorectal, lymphoma and leukaemia. With the recent exponential progress in drug development, programmed cell death, which includes apoptosis and autophagy, has become a promising therapeutic target. The research in identifying effective small molecules that target compensatory mechanisms in tumour cells alleviates the emergence of drug resistance. Due to the heterogenous nature of breast cancer, various attempts were made to overcome chemoresistance. Amongst breast cancers, triple negative breast cancer (TNBC) is of particular interest due to its heterogeneous nature in response to chemotherapy. TNBC represents approximately 15% of all breast tumours, however, and still has a poor prognosis. Unlike other breast tumours, signature targets lack for TNBCs, causing high morbidity and mortality. This review highlights several small molecules with promising preclinical data that target autophagy and apoptosis to induce cell death in TNBC cells.

## 1. Introduction

The early classification of cell death is into three main categories, apoptosis (type I), autophagy (type II) and necrosis (type III) [1]. However, in recent years, many new cell death modes of action have been developed based on the morphologies, molecular mechanisms and corresponding stimuli. This has led to the difficulties in incorporating these cell death modalities into the limited types I–III categories. Thus, by 2018, the Nomenclature Committee on Cell Death had listed multiple cell death types in a molecule-oriented manner [2]. Despite this, the interconnections among different cell death types still remain unclear, and their molecular interactions present further confusion.

Cell death may occur when the damage is severe, and the repair response is unable to provide protection or recovery, or it can be due to self-orchestration. Like other cell deaths, tumour cell death can occur programmed and accidentally, depending on the signalling pathway. Programmed cell death (PCD) closely depends on tightly regulated intracellular signal transduction pathways [3,4], whereas accidental cell death is referred to as non-PCD due to unexpected cell injury. PCD can be further divided into apoptotic cell death and non-apoptotic cell death based on morphological characteristics and molecular mechanisms (Figure 1). Apoptosis preserves cell membrane integrity and happens in a caspase-dependent manner. On the other hand, non-apoptosis cell death occurs by membrane rupture and it is caspase-independent.

Evasion of PCD is strongly associated with clinically aggressive cancers, which are characterised by early metastasis. Triple negative breast cancer (TNBC) refers to a subgroup of breast cancer that lacks oestrogen, progesterone and HER2 receptors and has a mortality rate of 40% within the first 5 years of diagnosis [5]. It was reported that 50% of patients were resistant to neoadjuvant chemotherapy [6]. Several factors are implicated in chemoresistance of TNBC which include cancer stem cells and the pathways that regulate cancer stem cells such as Hedgehog, Wnt/Beta Catenin, Notch and TGF-Beta, deregulation of the apoptotic machinery, alteration in the tumour microenvironment such as hypoxia, the role of ATP-binding cassette (ABC) transporters in drug efflux, microRNA and modulation of various signalling pathways, NF-kB, PTEN/PI3K/AKT/mTOR; JAK/STAT and receptor kinase [7]. Genotypic and phenotypic evolutions in response to chemotherapy have shown that TNBC cells harbour somatic mutations, a high frequency of p53 mutations and complex aneuploidy rearrangements which lead to intratumour heterogeneity [8]. The heterogeneous nature of TNBC results in failure of tumour biology-guided approaches in treating TNBC patients; however, targeted therapeutic approaches shows promising outcomes in clinical settings [9].

This review will highlight small molecules with promising preclinical data that target autophagy and apoptosis to induce cell death in TNBC cells. Here, we begin with the discussion of the recent preclinical research and developments of small molecules that affect the apoptotic, genomic instability, proliferating and stress-mediated signalling pathways of TNBC cells that lead to apoptosis. Next, we focus on small molecules that inhibit and enhance autophagy pathways in TNBC and other breast cancer cells through selectivity and sensitivity, and enhanced delivery methods. Finally, we summarise the perspective and potential of small molecules in promoting cell death through apoptosis and autophagy.

## 2. Apoptosis

Apoptosis was initially described as ‘shrinkage necrosis’ in the 1970s [10], which eventually marked the linkage between cell death and DNA degradation [11]. The progress in modern genetic techniques have made apoptosis one of the most well-studied modes of PCD, where the death signals culminate in the activation of caspases that lead to cell destruction (Figure 2). Apoptosis is triggered through two major pathways; an intrinsic and extrinsic pathway. The extrinsic pathway is triggered by death receptors (e.g., TNF), which are transmembrane proteins expressed on the cell surface, whilst the intrinsic pathway is mediated by mitochondria-released proteins (e.g., B-cell lymphoma 2 (Bcl-2) protein family). The morphological changes exhibited by apoptotic cells, which include cell blebbing and shrinkage, nuclear fragmentation, chromatin condensation and fragmentation and formation of apoptotic bodies, are well characterised and can be observed via various imaging techniques. Apoptosis as a potential therapeutic target in breast cancer was recently reviewed elsewhere [12].

Apoptosis, being one of the oldest PCDs studied, is a complex and complicated phenomenon that emanates from the interaction between various protein molecules and signalling pathways. The vast developments in high throughput study have provided substantial improvement in the development of small molecule compounds targeting specific pathways. A great number of studies have been conducted to develop molecules targeting the apoptotic pathway in breast cancer cells. In this review, small molecules that target pivotal signalling pathways involved in modulating hallmarks of cancer attributes, which include evading apoptosis, genomic instability, sustaining proliferative signalling and alteration in the tumour microenvironment, will be highlighted. In particular, emphasis is placed on data obtained in TNBC research.

### 2.1. Targeting Apoptotic Pathway

Caspases are cysteine proteases that serve as signalling mediators involved in orchestrating apoptotic execution pathways by proteolytically cleaving subsets of cellular proteins and are thus intimately involved in apoptosis [13]. The positive and negative regulators of caspases include the Bcl-2 family proteins, inhibitor of apoptosis (IAP) family, cytochrome c and second mitochondria-derived activator of caspases (SMAC), also known as Direct IAP-binding Protein with low pI (DIABLO) [14].

Multiple cancers were associated with high levels of IAPs and were shown to have poor prognoses. On the other hand, cancer patients with high levels of SMAC were shown to have longer overall survival [15]. SM-164 was previously reported as a water-soluble and cell-permeable bivalent SMAC-mimetic compound that binds to both the BIR2 and BIR3 domains of XIAP as well as cIAP-1 and cIAP-2 proteins [16]. The effects of SM-164 were previously investigated in head and neck squamous cell carcinomas and showed promising results as a potent radiosensitiser [17]. The group expanded their research to investigate if this effect was also observed in TNBC cells. SM-164 was shown to mediate caspase activation through rapid degradation of negative blockers specifically cIAP-1 and inhibit the binding of XIAP to active caspase-9 leading to activation of both intrinsic (SK-BR-3) and extrinsic (MDA-MB-468) apoptosis pathways [18].

TNBC patients presenting with loss of Bcl-2 expression were shown to respond better to chemotherapeutic treatment. This suggests Bcl2 as a promising prognostic and predictive marker in TNBC [19]. The authors used liposomes to incorporate Bcl-2-specific siRNA to therapeutically reduce the expression of Bcl-2 in TNBC cells. It was also reported that parenteral administration of nanoliposomes-Bcl-2 siRNA on MDA-MB-231 tumours in an orthotopic xenograft model in mice significantly reduced tumour growth. The nanoliposomes-Bcl-2 siRNA-treated mice showed marked induction of apoptosis, indicated by increased expressions of cleaved caspase-9 and poly(ADP-ribose) polymerase (PARP). The combination treatment between doxorubicin and nanoliposomes-Bcl-2 siRNA enhanced the tumour efficacy in TNBC in vivo model. Autophagy was also noted in tumours treated with nanoliposomes-Bcl-2 siRNA, indicated by enhanced expression of autophagy markers (LC-3II and ATG5) [20].

### 2.2. Targeting Signalling Pathways Involved in Genomic Instability

Genomic integrity in cells is maintained through DNA repair processes which are triggered as a response to DNA damage by to various genotoxic stresses. The DNA repair pathways are mainly classified as homologous recombination repair, non-homologous end joining repair and single-stranded break repair. PARP1 plays a vital role in DNA repair, which includes stabilization of DNA replication forks as well as altering its chromatin structure to facilitate the DNA repair response [21]. PARP inhibitors were shown to selectively targets cells with homologous recombinant deficiency, which are devoid of a homologous recombination repair pathway [22]. Several PARP inhibitors, including olaparib, niraparib, rucaparib and talazoparib, have been approved by FDA for cancer treatment.

BRCA mutations, specifically BRCA1, are more common among TNBC patients [23] and are closely associated with highly aggressive cancers and a poor prognosis [24]. The use of PARP inhibitor in TNBC cells results in impaired single-strand break repair mechanisms which ultimately lead to apoptosis [25]. The synthetic lethality interaction between BRCA and PARP1 has led to the development of several FDA-approved PARP inhibitors which showed encouraging outcome in TNBC patients. Hence, the small molecules developed to improve the current PARP1 inhibitors’ efficacy represent an extremely promising anti-cancer therapy.

Several studies attempted to develop compounds that extend the therapeutic applications of the current PARP1 inhibitors in TNBC patients, despite the BRCA1/2 mutation’s status. In this review, four PARP inhibitors with various mechanisms to elevate PARP-mediated apoptosis induction, such as enhancing PARP1 degradation, increasing the affinity to PARP1 binding and developing conjugates by modifying the linkers will be discussed.

Various hydrophobic tags such as adamantyl, fluorene, alkyl-substituted phenyl, diphenyl and dicyclohexyl were used to tag olaparib, and the ability of these tagged compounds to induce PARP1 degradation was evaluated in MDA-MB-231 cells. Compound 3a, which bears a fluorene moiety without a carbon spacer (Figure 3(Ai)), has been shown to most effectively induce PARP1 degradation compared with olaparib alone or other tagged compounds in the initial screening assay. It was reported that the length of the carbon spacer influenced degradation activity, with the shorter linker having the most efficiency. The docking experiment showed that compound 3a binds non-competitively to the active site of PARP1 in similar fashion to olaparib. PARP1 degradation by compound 3a was shown to induce proteasome-dependent apoptosis in unfolded protein response (UPR) and endoplasmic reticular (ER) stress-mediated cells indicated by upregulation of proapoptotic genes (Bax, p21, cytochrome c) accompanied by reduced expression of antiapoptotic gene, Bcl-2 (Figure 3(Aii)) [26].

Initially, the search for novel leading compounds was conducted by screening the chemical libraries of the Drugbank and ZINC databases, and the LibDock protocol was used to identify the top 500 hits. Candidate compounds that were eligible for co-crystallization screening were determined by CDOCKER protocol. The co-crystallization screening identified one potential lead compound from the Drugbank database (DB00321) named PA-10 (3-(10,11-dihydro-5*H*-dibenzo[a,d] [7]annulen-5-ylidene)-*N,N*-dimethylpropan-1-amine) which can bind to the nicotinamide pocket of PARP1 (PDB ID code 5HA9). The lead compound was further modified by changing the carbon chain’s length; increasing substituents containing hydrogen bond donors and altering the aromatic skeleton. OL-1 is a derivative of oxepin, via a four-step synthesis using isobenzofuran-1(3*H*)-one as the precursor (Figure 3(Bi)). OL-1 showed enhanced anti-proliferative effects in MDA-MB-436 cells compared with other breast cancer cells with the highest potency on PARP1 inhibition. It has good affinity to PARP1 binding, with two hydrogen bonds formed in GLY863 and 10 ns molecular dynamics simulations on OL-1/PARP1 complex showed OL-1 can steadily bind to PARP1. OL-1 was shown to induce apoptosis, indicated by morphological changes, increase in apoptotic cell ratio, upregulation of BAX and cleaved caspase-3, as well as downregulation of Bcl-2 expression (Figure 3(Bii)). Inhibition of cell migration was observed upon OL-1 treatment, suggesting OL-1 may inhibit metastases in TNBC cells. The in vivo results from the xenograft tumour model showed significant decrease in tumour volumes and tumour weights upon OL-1 treatment at high dose (25 mg/kg/d) with no toxicity effect, and no weight changes were observed in the liver, spleen or kidney. The in vivo antiproliferative and proapoptotic activities are in accordance with the results obtained from the in vitro assays [27].

PARP’s targeting of proteolysis-targeting chimeras (PROTACs) was developed using niraparib as the PARP1 binding moiety. The small molecule (Compound 3) was synthesized via an azide-alkyne Huisgen cycloaddition reaction to form a triazole moiety that connects the PARP1 and the E3 ligase ligands through a simple aliphatic carbon linker. It was shown to selectively induce significant ligand-binding and proteasome-dependent PARP1 cleavage in the MDA-MB-231 cell line, which eventually led to apoptosis, indicated by increased apoptotic cell ratio and upregulation of cleaved caspase-3 level. The exact mechanism of this small molecule warrants further investigation [28].

The ability of olaparib to inhibit PARP is contributed to by the piperazine ring located on the core, and substantial modification at this site alters its cytotoxic properties. Hence, the piperazine ring at the N-terminus was modified to generate the linker, and this molecule was referred to as PARPi (olaparib derivative). Different analogues were developed from the hybrid conjugate of BMS-001, a potent PD-L1 inhibitor and PARPi, using different spacers. Three conjugates were obtained, and all conjugates significantly induced early- to late-stage apoptosis and restored immunity by blocking the PD-1/PD-L1. Compound 3 showed the most potent synergistic cytotoxic activity in TNBC cells. It is generated from N-alkylation of the piperazine scaffold (of PARPi) with chloroethoxy ethoxy, followed by esterification with BMS-001. It was also found to significantly reduce the expression of cell surface PD-L1.

Molecular docking insights interaction experiments showed that the carboxamide portion of the phthalazinone functional group of the conjugate forms two hydrogen bonds with Arg865 (αJ) and Tyr896 (βd) in the nicotinamide-specific binding pocket of the PARP1 catalytic domain, which was stabilised by the additional π-π interactions in Tyr907. It forms hydrogen bond networks with histidine and arginine residues (His862 and Arg865, respectively), which are in the adenine ribose-binding (AD) pocket of the catalytic domain (Figure 3C). The interaction of the conjugate and BMS-001 within the binding cleft of PD-L1 dimer was mediated via hydrogen bonding, salt bridge and hydrophobic interactions. Furthermore, it forms hydrogen bonds between the O atom on the diethoxy-linker of the conjugate with the Arg113 residues in the binding pocket, as well as between the carbonyl in proxy to the piperazine ring with the Asp63 side chain. Other noncovalent interactions to stabilise the conjugate include π-π interactions with tyrosine residues in the binding cleft [29].

### 2.3. Targeting Pathways Involved in Proliferative Signalling

Unfavourable stress-induced conditions lead to arrest of the cell cycle transiently or irreversibly [30]. The uncontrollable cell proliferation observed in tumour cells is attained by either shortening the cell cycle or converting the resting or quiescent cells into proliferating cells [31]. Aberrant signalling from cell-cycle proteins promotes replication and division of a cell that leads to enhanced protein synthesis, which includes protein transcription and translation.

Eukaryotic elongation factor 2 kinase (eEF2K), which is a Ca^2+^/calmodulin-dependent kinase, is also a negative regulator of protein synthesis. Phosphorylation within the GTP-binding domain of eEF2 impedes recruitment of eEF2 to ribosomes, leading to impairment of ribosome elongation during mRNA translation [32]. eEF2K is regulated by several stimuli, including nutrient deprivation and energy depletion. Inadequate oxygen, as well as DNA damage, are involved in modulation of several cellular process, such as protein synthesis, cellular differentiation and tumourigenesis [33,34].

Tumour cells adaptation and survival to the microenvironment was shown to be modulated by eEF2K [35]. Several studies have suggested the role of eEF2K in tumour progression and chemoresistance in TNBC [36]. The two molecules discussed in this review either used the docking assay to identify lead compounds or utilised the currently available eEF2k inhibitor as a template to modify its structure and improve its selectivity in inducing apoptosis in TNBC cells.

eEF2K was previously reported as a potential therapeutic target in phosphatase and a tensin homolog (PTEN)/P53-deficient TNBC cells [37]. ChemBridge CORE library modified by Lipinski’s rule-of-five by Discovery Studio 3.5 was used in a three-step molecular docking screening to identify the lead compounds of eEF2K inhibitor. The hits were further screened using the LibDock protocol, followed by CDOCKER protocol that determines the hits against expression levels of eEF2K and p-eEF2K (ser78). Forty-six compounds were identified and compound 21l showed the most promising eEF2k enzymatic and anti-proliferative activities against TNBC cells. It was reported that the carboxyl group of 21l forms a hydrogen bond with the side chain of Try236 and an additional hydrogen bond between the cyano groups of 21l and Ile232. Both Try236 and Ile232 are residues located in the kinase hinge of the ATP binding site. Cyanophenyl can establish two π-π interactions with Try236 and Phe138. These allow stable binding of 21l to the ATP binding site of eEF2K, leading to potential ATP-competing inhibitor. Treatment of 21l against TNBC cells resulted in decreased anti-apoptotic protein (Bcl-2) and upregulated pro-apoptotic proteins (Bax and cleaved caspase-3, caspase-8 and PARP) as well as loss of mitochondrial membrane potential (Figure 4(Ai)). Apoptosis was visibly noted by morphological changes and a significant reduction in Ki-67-positive cells upon 21l treatment (Figure 4(Aii)). Tumour growth inhibition was also noted in xenograft mouse models of TNBC (MDA-MB-231 and MDA-MB-436) [38].

In improving the limitations of the current eEF2K inhibitors, PROTAC strategy was utilised, whereby it promoted eEF2K degradation and inhibited its autophosphorylation and downstream signalling. The analysis was compared to a known eEF2K inhibitor, A484954. A484954 was combined with the ATP binding pocket of eEF2K and formed hydrogen bonds with residues Lys170, Ile232, Gly234 and Try236. The authors tried using various lengths of linker and used thalidomide as a recruitment element. These modifications enhanced the degradation; however, not to a satisfactory level. Further modifications, which included a 3-OH substitution in thalidomide linked to *tert*-butyl bromoacetate, enhanced the degradation of ethyl-substituted eEF2K-PROTACs. One of the PROTAC compounds (11l) with the most carbonyl groups showed positive enhancement of the degradation. Compound 11l was able to significantly induce dose-dependent ubiquitin-proteasome-mediated eEF2K and p-eEF2K degradation and had a significant inhibitory effect on phosphorylated eEF2. Compound 11l showed significant upregulation of pro-apoptotic proteins (Bax and cleaved-caspase-3) and downregulation of Bcl-2 in TNBC cells (Figure 4B). Compound 11l exhibited anti-tumour effects by inducing apoptosis, which was further supported by imaging data [39].

### 2.4. Targeting Signalling Pathways Involved in Modulating Tumour Microenvironment

Cellular stress is an inherent characteristic of tumourigenesis and its response to this is associated with activation of proteins that contribute to tumour cells’ adaptation and survival, and promotes resistance towards chemotherapeutic drugs. Oxidative and ER stress imbalance were noted to augment different cellular functions, such as tumour proliferation, migration and differentiation. Hence, targeting these signalling pathways provides another emerging opportunity for cancer therapeutics.

#### 2.4.1. Endoplasmic Reticulum Stress

ER plays a pivotal role as a site for regulating protein folding and maturation in eukaryotic cells [41]. Disruption in ER functions, also known as ER stress, triggers UPR, which aims to restore protein function by the action of three signalling proteins named IRE1α (inositol-requiring protein-1α), PERK (protein kinase RNA (PKR)-like ER kinase), and ATF6 (activating transcription factor 6) [41]. However, prolonged stress leads to apoptosis via three primary pathways; IRE1/ASK1/JNK pathway, the caspase-12 kinase pathway and the C/EBP homologous protein (CHOP)/GADD153 pathway [42]. Collective studies in cancer have shown that PERK activation upregulates CHOP, a transcription factor involved in ER stressed-induced apoptosis [43].

Multiple cross-talks between the various pathways of ER-stress suggest the possible association of ER-stress with chemoresistance in tumours [44]. Activation of the ER stress pathways triggers pro-survival signals in tumours, which promote migration and invasion of cancer cells that characterize aggressive cancers [45]. Hence, targeting ER stress pathways shows potential therapeutic intervention in TNBC management.

Initially, various 5-nitrofuran-2-amide derivatives were shown to induce the expression of CHOP. Interestingly, SAR studies indicated that when six-ring substituents were introduced to the phenyl ring that forms the *N*-phenyl-5-nitrofuran-2-carboxamide skeleton exhibited a potent inhibitory effect on cell viability in TNBC. *N*-(4-iodophenyl)-5-nitrofuran-2-carboxamide derivative-treated HCC-1806 cells showed a drastic decrease of antiapoptotic proteins, such as Bcl-2 protein, and a significant increase in pro-apoptotic proteins; Bim, caspase-3, as well as cleaved caspase-4.

Another small molecule that was found to activate ATF4/CHOP-mediated ER stress response and induce growth inhibition in TNBC cells is ONC201. ONC201 was effective against TNBC cells with acquired TRAIL resistance; however, cells that were resistant to ONC201 were not responsive to TRAIL [46]. The group further explored the effects of ONC201 in a MDA-MB-468 xenograft model and reported tumour growth inhibition. The activation of the TRAIL-dependent apoptosis pathway by ONC201 led to in vivo antitumour efficacy. The TNBC cells with resistance to ONC201 required pRB to induce G1 cell cycle arrest, and a combination treatment with taxanes showed synergistic anticancer activity [46].

YM155 was identified as a novel survivin suppressor that induced apoptosis against TNBC cells and marked tumour regression in human TNBC xenograft tumours with minimal systemic toxicity. The group also reported YM155 treatment showed evidence of tumour metastasis inhibition and blocked tumour cell dissemination prior to the initiation of the metastasis process [47]. In a recent report, it was shown that the apoptotic effects of this small molecule are mediated via the ER-stress pathway. The therapeutic effect of YM155 in inducing apoptosis in TNBC models in combination with other anticancer drugs such as CD34-TRAIL1 was further explored. The pre-treatment of YM155 in TNBC cells before CD34-TRAIL1 treatment sensitizes the cells to both intrinsic and extrinsic caspase-mediated apoptotic pathways by upregulation of DR5 expression through a p38 mitogen-activated protein kinase (MAPK)- and CCAAT/enhancer-binding protein homologous protein (CHOP)-dependent mechanism [48]. YM155 has also been shown to induce cell death via autophagy, as explained in the following section.

YD277 is another small molecule involved in the ER-mediated apoptotic pathway, derived from ML264, a known Kruppel-like factor 5 (KLF5) inhibitor. KLF5 is frequently deleted in human cancer types and important in modulating various cellular processes, such as cell proliferation, apoptosis, migration and differentiation [49]. The authors developed several derivatives of ML264 and identified YD277 as the derivative with the most potent cytotoxic activity against TNBC cells. YD277 reduced the expression of Cyclin D1 and anti-apoptotic proteins (Bcl-2 and Bcl-xl) and induced cell cycle arrest by enhancing the expression of p21 and p27 (Figure 4C). It was also noted that YD277 activates ER stress-mediated apoptosis in TNBC cell lines by significantly inducing IRE1α transcription, which leads to JNK activation in a dose-dependent manner [50].

#### 2.4.2. Oxidative Stress

Reactive oxygen species (ROS) consist of hydroxyl (HO^*^) and superoxide (O_2_^*^) free radicals and non-radical molecules, such as hydrogen peroxide (H_2_O_2_). Elevated ROS levels that are accompanied with a decrease in cellular antioxidant enzyme systems promotes malignant transformation [51]. The cells are protected from oxidative stress by antioxidants which can be classified into three main groups; endogenous antioxidants (bilirubin, catalase, ferritin, superoxide dismutase (SOD), glutathione (GSH), coenzyme Q, l-carnitine, alpha lipoic acid, glutathione peroxidase (GPx), melatonin, metallothionein, thioredoxins, peroxiredoxins and uric acid), natural antioxidants (ascorbic acid, polyphenol metabolites, β-carotene, vitamin E, and vitamin A) and synthetic antioxidants (Nrf2, tiron, pyruvate, selenium and *N*-acetyl cysteine (NAC)) [52].

ROS-dependent apoptosis is achieved by either ROS induction or antioxidant inhibition, which will contribute to accumulation of ROS in cells and subsequently cancer cell death. Chemotherapy-induced ROS in cancer cells is a promising approach to induce selective cytotoxicity in cancer cells. Owing to the heterogenous nature of breast cancer cells, the success of this strategy requires characterisation of the redox status, which will help in enhancing the selectivity and efficacy of cancer therapy.

DCAC50, a small-molecule inhibitor of the intracellular copper chaperones, ATOX1 and CCS, was shown to elevate oxidative stress indicated by change in the GSH: oxidised glutathione (GSSG) ratio level and enhanced SOD activity in TNBC cells. The impaired intracellular copper transport alters the copper homeostasis resulting in decreased proliferation and enhanced apoptosis in TNBC cells (Figure 4D). Several TNBC cell lines with variable elevated levels of copper transporters (ATOX1 and CCS) were tested in an effort to recapitulate the tumour heterogeneity of breast cancers observed in clinical settings. DCAC50 showed antitumour effects more effectively in basal-like cell lines, such as MDA-MB-468 cells, which were noted to have lower baseline copper levels compared to claudin-low cell lines, which include MDA-MB-231, MDA-MB-436 and HCC1395. The efficacy of DCAC50 was investigated in the MDA-MB-468 xenograft mouse model, which showed inhibition of tumour growth and reduced tumour volumes upon DCAC50 treatment, with evidence of suppressed angiogenesis [40].

The authors used isotopic tandem orthogonal proteolysis-enabled activity-based protein profiling (isoTOP-ABPP) in screening 58 dichlorotriazine-based covalent ligands to identify druggable hotspots in TNBC cells, which led to the discovery of KEAI-97. isoTOP-ABPP is an advanced technology compared to ABPP, which uses isotopically tagged cleavable links in the iodoacetamide-alkyne probe to enable selective enrichment and release. The analysis was conducted using LC/LCMS/MS to quantify the light:heavy isotopic ratios [53]. KEA1–97 selectively impaired cell survival and proliferation of TNBC cells, which was evidenced by apoptosis induction mediated by protein-protein interaction disruption between thioredoxin and caspase-3, and targeting lysine 72 of thioredoxin [54].

### 2.5. Other SMALL Molecules

Several small molecules were reported to show early evidence of mediating the induction of apoptosis in TNBC cells. These small molecules were developed from various methods, namely the repurposing of old drugs derived from natural products or identified from screening assay.

Quinacrine was discovered during an intensive antimicrobial research; it is well documented for its use as an anti-malarial drug and has FDA approval for treating giardiasis and tapeworm infection [55]. The apoptotic effects of quinacrine in modulating cell cycle as well as nuclear fragmentation were noted in TNBC cells. Quinacrine induces S-phase arrest and a significant upregulation of p53, inhibitor cyclin-dependent kinase p21 (Cip1/Waf1), proapoptotic marker Bax and cleaved PARP product, and downregulation of antiapoptotic marker Bcl-xL [56].

The traditional uses of *Tinospora cordifolia* in jaundice, rheumatism, diabetes and inflammation, as well as the pharmacology activity in boosting the immune system and treating digestive disorders were previously reported [57]. A bioactive pyrrole, bis(2-ethylhexyl)-1H-pyrrole-3,4-dicarboxylate (TCCP) was isolated from the leaves of *Tinospora cordifolia* and shown to exert cytotoxic oxidative stress effects on TNBC cells. TNBC cells treated with TCCP showed elevated ROS and intracellular Ca^2+^ ion concentration. These events result in mitochondrial membrane potential dissipation and enhanced mitochondrial permeability transition pore formation which facilitates the release of cytochrome c, triggering apoptosis by the activation of caspase-3 and 9. Interestingly, TCCP effects were also tested in the murine ascites carcinoma model, which showed a significant reduction in tumour burden with no toxicity observed in either renal or liver cells [58].

The small molecule pyridopyrimidinone inhibitor, FRAX1036, was derived from a previously optimised PAK inhibitor, FRAX597. FRAX1036 does not contain the characteristic arylamino moiety, which resulted in improved kinase selectivity and pharmacokinetic profile. A combination treatment of docetaxel and FRAX1036 was shown to enhance apoptosis, with increased PARP cleavage and attenuated cell cycle regulator (cyclin D1) noted in the high content time-lapse imaging used to monitor apoptosis [59]. The combination treatment hastened the docetaxel-induced mitotic arrest, causing microtubule disorganisation that lead to kinetic enhancement of breast tumour cell apoptosis.

Sirtinol is a SIRT1 inhibitor, found to induce p53-mediated apoptosis, caspase-independent autophagy and G1 cell cycle arrest in MCF-7 cells [60]. (E)-*N*-Benzyl-2-[(E)-[(2-hydroxynaphthalen-1-yl)methylidene]amino]-4,5,6,7-tetrahydro-1-benzothiophene-3-carboxamide (JGB1741) is a small molecule inhibitor of SIRT1 that was shown to have better cytotoxic activity than Sirtinol. As compared with Sirtinol, the benzene of 3-amino-benzamide of Sirtinol was replaced with thiophene in JGB1741. Due to the absence of SIRT1’s crystal structure, JGB1741 was generated using the best model of the catalytic core of SIRT1 and the fitness of the model was checked by the PROCHECK program. JGB1741 was shown to elevate acetylated p53 levels, leading to p53-mediated apoptosis with the regulation of Bax/Bcl2 ratio, cytochrome c release and PARP cleavage. It is also potent in inhibiting cell proliferation, mainly in metastatic TNBC cells [61].

The epigenetic regulator enhancer of zeste homolog 2 (EZH2) that is usually overexpressed in many cancers is the catalytic subunit of polycomb repressive complex 2 (PRC2), functioning as a methyltransferase to induce the trimethylation of histone H3 at Lys27. It was shown to play a pertinent role in tumour plasticity, which is required for cancer cells’ adaptation to the tumour microenvironment [62]. ZLD1039 is a potent and selective inhibitor of EZH2, designed and synthesized from pyridone-containing chemical scaffold compounds using classical a medicinal chemistry approach to identify EZH2 small molecule inhibitors. ZLD1039 was shown to inhibit EZH2 enzyme activity and upregulate CDK inhibition, which lead to cell cycle arrest and apoptosis. These effects were also observed in human breast tumour xenograft models. The antimetastatic activity of ZLD1039 was postulated due to the recovery of CDH1 (E-cadherin) and associated with matrix metalloproteinase (MMP)-2 and MMP-9 [63].

Separase belongs to the CD clan of cysteine proteases and it regulates and controls the onset of separation of sister chromatids in anaphase during mitosis in all eukaryotes [64]. The compound 2,2-dimethyl-5-nitro-2*H*-benzimidazole-1,3 dioxide, also known as Sepin-,1 is a separase inhibitor identified from high-throughput screening of a small molecule compound library, using fluorogenic separase assay. It was shown to possess anti-apoptotic and anti-proliferative activities in in vitro assays. Sepin-1 activity against patient-derived xenografts from two TNBC (MCI and BCM-547) showed mixed results, wherein Sepin-1 treatment inhibited the growth of xenograft in MCI tumour, whilst no significant difference was observed between vehicle control and Sepin-1 treatment in BCM-547. The authors suggest that the level of separase effects Sepin-1 sensitivity because MC1 tumours have higher separase levels compared with BCM-547 [65].

Overexpression of MDM2 and NFAT1 in cancers is correlated with poor prognosis and inhibition of these proteins is a promising targeting strategy for cancer therapy [66]. NFAT1 modulates MDM2 transcription and activates MDM2 expression, which leads to an increase in p53 inactivation that drives tumourigenesis. The Zhang lab initially identified genistein as an MDM2 and NFAT1 dual inhibitor. They expanded their search to develop small molecules with more potent activity, using a computational structure-based screening of natural products that led to the discovery of JapA, LinA and InuA [66]. A recent report on LinA showed that it is less sensitive in inducing apoptosis in TNBC cells, hence will not be discussed in this review. InuA selectively suppressed cell proliferation by causing cell cycle arrest at the G2/M phase and induced apoptosis in TNBC cells in p53-independent manners. The effect was also observed in mice with orthopedic tumours without toxicity. Inhibition in cell migration and invasion were observed in in vitro assay upon InuA treatment and this was reciprocated in animal models, where InuA suppressed tumour metastasis [67].

## 3. Autophagy

Autophagy allows cells under stress to regain control of their damage by a monitored delivery of cellular materials to lysosomes for recycling and degradation [68]. Mitochondria regulate autophagy by way of the limited number of autophagy-related genes (ATGs) [69] and production of ROS [70]. On the contrary, autophagy controls mitochondrial homeostasis through mitophagy [71]. This unique relationship between autophagy and mitochondria dictates many discoveries in this field of research. Three forms of autophagy have been identified, macro-, micro- and chaperon-mediated autophagy. Among these, macroautophagy is a major regulated catabolic mechanism that involves the formation of a compartment called autophagosome that seizes and transports its content for controlled fusion with lysosome (Figure 5) [72,73]. This act can either lead to the death or survival of the cell and is dependent both on the tissue and the microenvironment [74,75,76,77,78].

Autophagy is a survival pathway of cancer cells exposed to genotoxic stress and activated by oncogenic signals. Many oncogenic proteins, such as class I phosphatidylinositol 3-kinase (PI3KCI), protein kinase B (AKT), mammalian target of rapamycin (mTOR), Bcl-2 and mitogen-activated protein kinases (MAPKs), may suppress autophagy, while others, including PI3KCIII, PTEN, Death-Associated Protein Kinases (DAPKs), Beclin-1, Bax-interacting factor 1 (Bif-1) and p53 (depending on the type of cancer), promote it [79,80,81]. In many advanced cancers, autophagy facilitates tumour growth by promoting adaptation to environmental and metabolic stress [77,82]. Several in vivo and in vitro studies have demonstrated that autophagy contributes to the development of chemo- and radiotherapy resistance [75].

Most small molecules are designed to inhibit or enhance autophagy pathways in TNBC and other cancer cells. The promising investigations presented here promote the development of drug combination targeted therapy and enhanced drug delivery in TNBC. As a result, much more is yet to be learned from the discovery of novel small molecules that are sensitive and can selectively increase (or decrease) autophagy-dependent cell death. This ultimately will deliver effective breast cancer therapeutics.

A drug that can selectively target tumour cells is highly desirable in cancer treatment. Cancer involves multiple pathways with many protein receptors that are intertwined in their roles, thus creating a complex disease. These protein receptors are usually targeted to cause cell death, including autophagy. Although a lot of progress in this area of research has been made, many protein receptors’ functions are yet to be fully understood. Here, we would like to highlight some of the small molecules that are selective and sensitive towards inducing and inhibiting autophagy in TNBC and other breast cancer cells.

### 3.1. Synthetic Small Molecules as Autophagic Inducers

Breast cancer is histologically diverse, with various molecular subtypes of TNBC, and has a complex tumour profile and aggressive behaviour with high rates of reoccurrence. TNBCs are driven by multiple signalling pathways involving several kinases. Zhou et al. have identified KIN-281, a quinazoline-based small molecule, to inhibit various kinases including maternal embryonic leucine zipper kinase (MELK) and bone marrow X-linked (BMX). With the aim of discovering new therapeutic targets, a structure-based molecular docking screening was conducted to identify compounds that inhibit MELK. MELK overexpression is associated with poor prognosis in cancer as it promotes cancer cell survival and has strong implications for regulating cell cycles by causing G2/M arrest [83,84,85]. KIN-281 was shown to inhibit several kinases in TNBC cells, including MELK, FER kinase, BMX/ETK and TIE2/TEK, with IC_50_ values ranging from 1–4 mM to 42.7 mM. An upregulation of p21 WA/CIP1, corresponding to a decreased level of cyclin A2, was observed upon KIN-281 treatment. This was postulated to be associated with inhibition of p53 phosphorylation at Ser15. KIN-281 was also shown to inhibit transcription 3 (STAT3) phosphorylation but, interestingly, suppressed expression of pro-apoptotic proteins was reported. This prompted the investigators to explore the effect of KIN-281 on autophagy, in which they concluded was an inhibition of autophagy flux upon KIN-281 treatment in TNBC cells (Figure 6A) [86].

As mentioned earlier, the prognosis for TNBC is poor, however Vogel and co-workers have shown that piperidine-based small molecules, b-AP15 and RA-9, are able to inhibit deubiquitinating enzymes (DUBs), and this has a significant effect on TNBC cell viability. The small molecules interact with DUB, which leads to autophagic activation to compensate for ubiquitin-proteasome-system (UPS) stress rather than cellular death. Their research was expanded to show that this phenomenon is a common feature for other cancer types, including ovarian cancer. They are optimistic that, with the combination of these small molecules and other autophagy-inducing molecules such as vorinostat or chloroquine, a treatment for TNBC will be possible [87].

Chang and co-workers have shown that the selenium-based small molecule, SLLN-15 is able to selectively induce cyctostatic autophagy in TNBC cell lines such as MDA-MB-231 and BT-20. High-throughput biochemical enzymatic assays were used on a library of seleno-purines that identified SLLN-15, which selectively inhibits AKT-mTOR signalling, decreasing AURKA expression (91% inhibition) and Janus kinase 2 (85% inhibition). In vivo studies, using 30 mg/kg of SLLN-15, reveal that this small molecule is a potent anti-cancer drug and shows anti-metastatic activity in mice with TNBC [88].

An extensive study exploring interactions at active sites, distances of target proteins and database search was developed to synthesise a pharmacophore model of mTOR inhibitors to induce autophagy in TNBC cells. They also looked at the best interaction energies and scores according to the Lipinski’s rule-of-five drug-like principles, using the LibDock and CDOCKER protocols. A lead compound, 3-bromo-*N*’-(4-hydroxybenzylidene)-4-methylbenzohydrazide, was identified and further structurally modified, yielding a compound with inhibition of mTOR with IC_50_ of 0.304 μM [89].

Cancer cell metastasis treatment is also achieved by inhibition of Aurora A (AURKA) kinase, resulting in activation of the LC3B/p62 axis and inhibition of pAKT, thus inducing cytotoxic autophagy. Kozyreva et al. targeted the inhibition of AURKA by using a combination treatment of small molecules, MLN8237 and eribulin on different stages of metastasis in TNBC, resulting in suppression of metastatic colonisation and recurrence of cancer with cytotoxic autophagy [92].

Another attempt to modulate cell death in TNBC was investigated in great length by the Liu group. In 2015, Zhang and co-workers explored the autophagy-related protein 4B(ATG4B) and identified flubendazole as a small molecule that targets this protein and causes anti-proliferative efficacy in MDA-MB-231 cells [93]. Two years later, they again applied extensive in silico high-throughput screening, kinase and anti-proliferation activity screening, site mutagenesis and chemical synthesis to build a series of small molecules which were able to act as agonists for unc-51-like kinase 1 (ULK1), a key regulator of autophagy initiation. The best candidate, LYN-1604, consists of a piperazine core with 3 large and generally hydrophobic groups (Figure 6B). The *sec*-butyls and phenyl moieties are observed to form hydrophobic interactions with the leucine and tyrosine residues and hydrogen bonds are formed by the ether oxygen with the lysine residue found in ULK1. These interactions give favourable binding energies and reasonable orientations of the small molecule to activate ULK1 with EC_50_ (concentration to reach 50% of the maximal activation) of 19 nM, revealing it as a potent agonist. Its autophagic cell death mechanism involves the activation of ULK1, ULK complexes and other interactors, such as autophagy related 5 (ATG5), activating transcription factor 3 (ATF3), double-strand break repair protein rad21 homolog (RAD21) and caspase-3, and provides promising understanding for the development of future TNBC therapeutics [90].

p53 regulates the Granzyme B apoptosis pathway, which is highly dependent upon eliminating tumour cells that are mediated by natural killer cells (NK) and cytotoxic T lymphocytes (CTL). Chollat-Namy et al. worked on reactivating p53-mutated tumour cells with a small molecule, CP-31398, a quinazoline-based compound, to increase its sensitivity to CTL or NK lysis. Treatment of breast cancer cells (MDA-MB231) with this small molecule for 24–48 h resulted in enhanced and reduced expression of several genes belonging to the p53 pathways. This small molecule induced autophagy in a p53-dependent manner via the induction of Sestrin-1 and -2 expressions, resulting in AMPK activation and mTOR inhibition and induction of ULK1. It was also noted that in their model, they did not observe any significant ROS production, which can induce autophagy after treatment with the small molecule, contrary to previous reports [94,95,96].

There are several studies using both TNBC and other cancer cell lines, as discussed further here. Sun et al. investigated the use of the small molecule fluoxetine in inducing autophagy in cancer cells. Fluoxetine was able to decrease cell proliferation and induce autophagic cell death by inhibiting eEF2K and activating the 5′-AMP-activated protein kinase-mammalian target of rapamycin-Unc-51-like autophagy activating kinase (AMPK-mTOR-ULK) complex in breast cancer cell lines, especially MDA-MB-231 and MDA-MB-436. They confirmed these mechanisms of action, as they recorded a time-dependent elevation of microtubule-associated protein 1 light chain 3-II (LC3-II) and the expression of Beclin-1, both of which are biomarkers of autophagy; increased percentage of cells in sub-G1 phase and up-regulated levels of caspase-3/8 and PARP level were also recorded, both are indicative of cell death [97].

Survivin is upregulated in many tumour cells and is closely related to metastatic spread, tumour invasiveness and poor prognosis in relation to treatment resistance. Its level, however, is barely detectable in normal cells, making survivin an attractive target for cancer therapy intervention [98,99]. YM155, a heterocyclic small molecule is identified as the first drug to inhibit survivin expression and has passed phase two clinical trials for various kinds of cancers [100]. However, its exact mode of action is still rather unclear. A study looked at this very subject and revealed that YM155 induces cell death in vitro with IC_50_ at nanomolar concentrations (40 nM for MCF-7, 50 nM for MDA-MB231 and 70 nM for Cal-51) via autophagy in a p53-independent pathway, despite recording DNA damage. They also discovered, using an ex vivo model, that the canonical NF-kB pathway potently regulates YM155-induced cell death via exponentially promoting the autophagy process. These results indicate that using YM155 in combination with other anti-cancer drugs is a concern, as multiple pathways are interfered with by this small molecule [101]. Interestingly, Cheng et al. studied the molecular mechanism of YM155 in various drug-resistant breast cancers, such as tamoxifen-resistant and caspase-3 deficient breast cancers, revealing IC_50_ values in the low nanomolar range. YM155 modulated autophagy and induced autophagy-dependent caspase-7 activation and autophagy-dependent DNA damage in cancer cells [102].

In another study, basal-like breast cancer (BLBC) was the focus of investigation. The growth and activity of BLBC are governed by N-Ras (neuroblastoma Ras) proteins, mainly to mediate growth factor signalling at the plasma membrane. It has always been a challenge to directly target these proteins in treating cancer [103,104,105]. However, Zheng and co-workers, through extensive screening, managed to identify flunarizine as a candidate to induce N-Ras degradation. Flunarizine is an FDA approved drug for the treatment of epilepsy and migraine. This piperazine-based small molecule can silence the N-Ras activity and selectively inhibit the growth of BLBC cells in vitro and in vivo, but not that of breast cancer cells of other subtypes, thus changing the autophagy pathway of these cancer cells [106].

There are multiple other small molecules, identified, through various screening techniques, as autophagic inducers in cell lines other than those found in TNBC cells. Some of these small molecules are used in combination with other known drugs to induce autophagy. Dragowska and co-workers investigated alteration of the cellular processes of autophagy in breast cancer cells induced by a small molecule called gefitinib. This compound has a quinazoline core and is known for its ability to inhibit epidermal growth factor receptor (EGFR) tyrosine kinase [107,108,109]. It was able to induce appearance of vesicular organelles in the cytoplasm regardless of the sensitivity of the cell towards the small molecule. This observation was recorded using imaging techniques such as high content analysis (HCA) and transmission electron microscopy (TEM). Treatment of cancer cells with gefitinib, in the presence of lysosomotropic agents such as hydroxychloroquine (HCQ) and bafilomycin A1 (which inhibits late-stage autophagy), increased the efficacy of gefitinib in vitro and in vivo. Combination treatment with HCQ in vivo gave 58% reduction in tumour volume in gefitinib-insensitive cells. This result was shown to be highly dependent on the lysosomotropic agents’ concentrations and the reversibility of the autophagy upon removal of the drugs [110].

In breast cancer cells, human epidermal growth factor receptor-2 (HER2) modulates tumour cell growth and survival. Tumours rely on the signalling of HER2, which activates type 1 phosphoinositide 3-kinase (PI3K) [111]. Young and co-workers have targeted type I (p110α) and type III (vacuolar protein sorting 34, Vps34) PI3Ks using SAR405, a pyrimidine-based small molecule, to inhibit these kinases to reduce tumour growth and induce autophagy. They have also shown that a locked nucleic acid antisense oligonucleotide (LNA-ASO), called EZN4150, is also capable of sensitising HER2; however it does not induce autophagy, and even blocks autophagy induction, in response to catalytic type I PI3K inhibitors. The small molecule and the oligomer, in combination, are assumed to inhibit Vps34-dependent pathways via tumour cell apoptosis [112].

Bromodomain-containing protein (BRD4) plays a key part in the development of many types of cancers, as it is related to oncogenic rearrangements, breast cancer cell metastasis regulation through modulation of enzymatic activity and extracellular matrix gene expression [113,114,115,116]. Ouyang et al. have shown that BRD4 may interact with AMP-activated protein kinase (AMPK). Inhibition of BRD4 by a small molecule was investigated by using molecular docking software, wherein they were able to narrow the search to one compound, from 500 hits. The identified compound, FL-411, is able to induce ATG5-dependent autophagy-associated cell death by inhibiting BRD4-AMPK interaction, resulting in the activation of the AMPK-mTOR-ULK1-modulated autophagy pathway in breast tumours. BRD-FL-411 interactions essentially consist of hydrogen bonding by the OH with Asn140, the methyl groups occupying the small hydrophobic pockets, and π-π interaction between the pyrimidinone with the indole side chain of Trp81 (Figure 6C). Other interactions, such as two hydrogen bonds by the amide scaffold of residues Pro82 and Gln85 with the hydrogen donor and acceptor, respectively, of the small molecule. Additionally, two water molecules stabilise FL-411, with one water molecule forming bridged hydrogen bonding with Asp88 and Pro86, and the other with Tyr97. These interactions result in the strong binding affinity of FL-411, making it a potent and selective BRD4 inhibitor [91].

### 3.2. Small Molecules from Natural Products as Autophagic Inducer

For many centuries, natural products have successfully been a source of drug development for tumour treatment [117,118,119]. Sulforaphane is a natural product, bearing an isothiocyanate functional group that displays potent anti-cancer activity against various tumour types [120,121,122]. Yang et al. have shown that sulforaphane induces membrane translocation and acetylation modification of phosphatase and tensin homolog (PTEN) as the underlying mechanism of sulforaphane in the inhibition of histone deacetylase (HDAC), via autophagy. This was demonstrated using TNBC cell lines, MDA-MB-231 (IC_50_ 21.8 μM), BT549 (IC_50_ 20.5 μM) and MDA-MB-468 (IC_50_ 21.9 μM). They have also shown that at low-dose combination of sulforaphane (50 mg/kg) and doxorubicin (2 mg/kg) synergistically inhibits cell growth [123]. They have previously shown that sulforaphane is able to downregulate HDAC5 gene expression. This discovery is important as high expression of HDAC5 inhibits lysine-specific demethylase 1 (LSD1) promotes TNBC cell proliferation and migration [124,125].

To increase the therapeutic sensitivity of TNBC, several small molecule inhibitors of ubiquitin enzymes were identified by Huang et al. [126]. Ubiquitin enzymes, in particular ubiquitin-conjugated enzymes (E2s), have a role in breast cancer progression and are suggested to increase therapeutic resistance in TNBC [127,128]. In the current study, they developed a fluorescence resonance energy transfer (FRET) assay based on UbcH5b-HECTD3 auto-ubiquitination. The assay contained enzymes and fluorophores, and upon catalytic reaction, two fluorophores came in close proximity to produce a FRET signal. However, in the presence of an inhibitor, this signal was blocked. Using this assay, they screened 291 natural products to identify three triterpenoids that were able to block the FRET signal and inhibit auto-ubiquitination, in vitro and in vivo, at 5–20 μM concentrations. Also, in the presence of the triterpenoids, when the ubiquitin enzymes were inhibited, the sensitivity of lapatinib towards TNBC in vitro was increased.

Small molecule, *N*-desmethyldauricine, LP-4 is another natural product that has shown to induce autophagy in other cancer cell lines. LP-4 is an alkaloid isolated from a grape-like fruit plant called *Menispermum dauricum* DC. Concentrations as low as IC_50_ 15.5 μM (MCF-7) and 19.7 μM (A549) are needed to induce autophagic cell death. LP-4 is able to do so via the ULK-1-PERK and Ca^2+^/Calmodulin-dependent protein kinase β (CaMKKβ)-AMPK-mTOR signalling cascades. The isoquinoline-based small molecule can mobilise calcium release via inhibition of SERCA, and cause cell death in a broad spectrum of cancer cells, both apoptosis-defective and apoptosis-resistant. A combination of tools was used in this study, including computational docking analysis and biochemical assays. Other natural alkaloid small molecules are also capable of inducing autophagy by stimulating AMPK-mTOR, such as liensinine, isoliensinine, cepharanthine and dauricine. However, this study shows the first mechanistic understanding of the role of cytosolic calcium levels in autophagic cytotoxicity [129].

Steroid receptor coactivators (SRC) can drive target gene expression via interaction with nuclear receptors and other transcription factors. They also function in various physiological processes, including growth, reproduction and metabolism [130]. Overexpression of SRC simulates a family of oncogenes in breast cancer, and this relates to large tumour size and poor survival rate [131]. Wang et al. took advantage of this pathway to cause acute super-activation of SRC coactivators using a pyridine-based small molecule, MCB-613, to effectively kill cancer cells by inducing cellular stress. MCB-613 is also categorised as a member of the chalcone family, whose members are intermediates in the biosynthesis of flavonoids in plants. The direct binding of MCB-613 to the receptor-interacting domain (RID) of SRC is reversible; the binding phenomenon can be monitored by an increase in signal on surface plasmon resonance (SPR) and a decrease in fluorescence emission of the small molecule. Thus, the overexpression of SRC could possibly be a strategy to kill tumour cells [132].

### 3.3. Small Molecules as Autophagic Inhibitors

Thus far, we have discussed small molecules, developed for the purposes of autophagic induction, from synthetic small molecules and from natural product sources. In this section, several small molecules have seen their way to inhibiting autophagy, which is another strategy in cancer treatment. A combination of several drug molecules may provide a novel therapeutic strategy, especially through the inhibition of autophagy [133,134,135,136]. Wang and co-workers have shown that dichloroacetate could significantly inhibit autophagy induced by doxorubicin in breast cancer cells (MDB-231) in vivo and prolong mouse survival. Dichloroacetate could inhibit the enzyme pyruvate dehydrogenase kinase, which changes the mitochondrial respiration of cancer cells and so causes apoptosis. Dichloroacetate also enhances doxorubicin-induced cell death and anti-proliferation in vitro, however its efficacy is significantly hampered by rapamycin (autophagy accelerant) treatment. These findings suggest that dichloroacetate is a potential small molecule that can be utilised in combination with other classic chemotherapeutic drugs for breast cancer treatment [137].

Autophagy has also been documented to contribute to the development of resistance to cancer treatments [75,138]. Vps34 is a PI3P kinase fundamental to the biogenesis of autophagosomes. A small molecule, SB02024, whose structure has not been disclosed, is able to effectively inhibit Vps34 with K_d_ of 1 nM and consequently inhibit autophagy and cell viability of breast cancer cells, both in vitro and in vivo. Moreover, in combination with Sunitinib, which is an indolinone-based, FDA-approved drug, SB02024 is able to enhance the cytotoxic effects of this drug. Two breast cancer cell lines, MDA-MB-231 and MCF-7, show increased sensitivity to Sunitinib in vitro in the presence of SB02024. This is not the first example of combinations of Vps34 inhibitors with anti-cancer drugs that have been reported [112,139,140,141]; however this study strengthens the finding that Vps34 can be a target for anti-cancer treatment [142].

In the effort of finding other autophagic inhibitors, Carew and co-workers targeted lucanthone, a tricyclic compound which has been previously explored as anti-schistosome agent [143,144]. In this study, they have recorded reduced cell viability in seven breast cancer cell lines with an impressive mean potency of IC_50_, 7.2 μM compared to chloroquine, 66 μM. Lucanthone inhibits autophagy and induces apoptosis in cancer cells through a p53-independent mechanism, mediated by an increase in cathepsin D, matrix metalloproteinase-1 (MMP1) and cytochrome P450 (CYP1A1) levels. They observed no signs of drug-related ocular toxicity, as recorded for chloroquine, or HCQ. Their results show the potential of this drug used in combination with others for chemo- and radiotherapy [145].

Cosford, Shaw and co-workers introduced SBI-0206965, which is a pyrimidine-based small molecule that inhibits ULK1 with an in vitro IC_50_ of 108 nM [146]. Through extensive molecular docking studies, they have determined the suitable moieties to attach to the next general class of compounds. Recently, they developed the more potent, SBP-7455, that binds to ULK1/2 and inhibits their enzymatic activity in vitro and in vivo. Thus, this resulted in reduced viability of TNBC cells through autophagy and exhibited favourable pharmacokinetic and pharmacodynamic properties. In combination with the PARP inhibitor olaparib, SBP-7455 synergistically kills MDA-MB-468 TNBC cells, displaying a more effective mode of action through PARP and autophagy inhibition to kill tumour cells than either mechanism alone [147].

The powerhouse of the cell, mitochondria, has a close relationship with autophagic behaviour, as described in the introduction [70]. As a result of this relationship, modulation of autophagy through modulating mitochondrial function by interaction with small molecules is an extremely interesting field of study. Aumitin is a pyrimidine-based autophagy inhibitor that targets mitochondrial respiration complex 1. Although aumitin is able to induce ROS formation, this species is not found to be responsible for autophagy inhibition. On the other hand, aumitin alone is able to inhibit starvation- and rapamycin-induced autophagy dose-dependently, which is suggestive of a downstream pathway target of mammalian target of rapamycin (mTOR) [148].

### 3.4. Enhance Delivery

Nano delivery and nanomedicine have become a rapidly growing field since materials science, especially at the nano scale, has seen tremendous development. Materials in the nanoscale range are used to deliver therapeutic agents or as diagnostic tools to target specific sites. This valuable approach has seen its application in the treatment of various diseases, including breast cancer, as discussed in this section.

Metal-centred nanoparticles have been exploited to target chemo-resistant breast cancer cells, highlighting the significance of nanomedicine in drug carrier technology [149,150]. The Shekhar group has extended their work on a triazine-based small molecule, SMI#9 [151], which targets Rad6, a protein that is overexpressed in aggressive breast cancer and involved in DNA damage tolerance, into nanoparticle chemistry [152,153,154]. They found that SMI#9 selectively induced cancer cell toxicity; however, it displays inferior efficacy due to poor solubility. To overcome this limitation, they have covalently attached SMI#9 to gold nanoparticles via ester bonds (Figure 7A), forming a superior material in the size range of 32–40 nm, that is able to undergo endocytosis and elicit cytotoxicity selectively in mesenchymal TNBC cells. The released the SMI#9 molecule via hydrolysis of the ester bonds in the cell, inducing cell death through mitochondrial dysfunction and PARP-1 stabilization/hyperactivation. In combination with cisplatin, this nanoparticle synergistically increases cisplatin sensitivity with GI_50_ by five-fold, recording 3.8 μM (MDA-MB-468) and >25 μM (HCC1937) using cisplatin alone compared with 0.8 μM (MDA-MB-468) and 4.9 μM (HCC1937) using cisplatin/nanoparticles. Their results suggest that the SMI#9-gold nanoparticles are able to target Rad6 for TNBCs therapy [155].

There is much literature on improved breast cancer patient outcomes through the targeting of inhibition of mTOR kinase, which exists as two structurally and functionally distinct complexes, mTORC1 and mTORC2 [157]. Classic chemotherapeutic drugs, such as rapamycin-related drugs, are found to be extremely sensitive to mTORC1 but, however, insensitive to mTORC2 and, in some cases, the over-use of these drugs are found to impair mTORC2 signalling [158]. There is also evidence, from preclinical and clinical genetic studies, to suggest that inhibition of mTORC2 while sparing mTORC1 signalling is highly desirable [159,160,161]. Werfel et al. have demonstrated the selective inhibition of mTORC2 in a preclinical model of breast cancer using nanoparticles delivered intratumourally and intravenously. The nanoparticles are composed of three components, siRNA, co-polymer (DB and PDB) and a polyethylene glycol-base, forming a corona structure (Figure 7B). They have targeted the gene *Rictor* and formulated Rictor-RNAi (RNA interference), which selectively inhibits mTORC2. More importantly, this method was also effective in TNBC models by decreasing Akt phosphorylation and tumour growth [156].

There are several groups interested in the progression of autophagy, thus investigating the relevant peptides involved. Beclin-1 is a peptide that participates in the regulation of autophagy via binding to PI3Ks, which are important in the initiation process of autophagosome formation in autophagy [162,163]. Thus, the overexpression of Beclin-1 could inhibit tumour development [134,164,165,166]. Wang and co-workers have designed autophagy-inducing peptides into polymeric nanoparticles to significantly induce autophagy and interfere with breast cancer cells in vitro and in vivo. Beclin-1 was covalently attached through the N-terminal onto poly(β-amino ester), which is a pH-sensitive polymer. This co-polymer was then self-assembled into micellar-like nanoparticles with poly(ethylene glycol) as the hydrophilic shell. This micellar structure is stable and thus has an enhanced uptake of Beclin-1 by endocytosis in vitro. The nanoparticles can induce lysosomal dysfunction by increasing the lysosomal pH and this results in the disassociation of the nanoparticles, releasing the protein and effectively inducing autophagic cell death. Additionally, Beclin-1 polymer inhibits tumour growth and retards in vivo tumour development. This nanomaterial is suggested to overcome the limitations of therapeutic Beclin-1 peptide, such as in vivo chemical instability and non-specific bio-distribution in tissues [167].

## 4. Conclusions

The structures and targets of more than 40 small molecules highlighted in this review is summarised in Table 1. Preclinical drug development uses both in vitro and in vivo assays to evaluate the biological and pharmacological activity of the clinical candidate molecules. The rapid advancement experienced in biotechnology, bioinformatics and nanotechnology has significantly improved drug discovery programs in the recent years. High-throughput screenings, coupled with in silico approaches such as molecular docking studies, allow rapid screening of compounds to identify potential leads and provide several advantages, including shorter times and financial investments in the drug discovery process. Difficult protein targets, such as driver proteins, are often termed as ‘undruggable’ targets that conventional drug development fails to target [168]. Proteolysis approaches, such as PROTAC, that target protein degradation and ultimately cause PCD, represent an alternative strategy for these ‘undruggable’ targets, such as PARP1 and eEF2K, which were highlighted in this review. PROTAC often represents the chemical equivalent of small interfering RNA (siRNA), which also causes proteolysis; however, it allows protein removal at the post-translational level, as compared to silencing at the post-transcription level in siRNA. The relatively new, but rapidly developing, field of nanomedicine is shown to improve pharmaceutical indices of active products, mainly drugs that are engineered into nanoparticles.

The main goal of cancer therapy, since the 1990s, when PCD became a therapeutic strategy, has been to selectively eliminate tumour cells. Initially, apoptosis, as a drug target, gained tremendous support from the scientific community, as pivotal knowledge was gained on the mechanism leading to the cell death response. Although the concept of autophagy was coined in the 1960s, the ‘double-edged sword’ role played by autophagy in tumour progression continues to serve as a promising tool for anticancer therapy. This review highlighted the small molecules that are designed to be selective and sensitive towards PCD pathways—mainly apoptosis and autophagy—in TNBC and other breast cancer cell lines. Autophagy and apoptosis display complex relationship in inducing cell death, and the preclinical data on small molecules targeting PCD shows encouraging progress. The emerging multidisciplinary approach in identifying lead compounds and target validation provides potential therapeutic strategies in treating breast cancer.

## Figures and Tables

**Figure 1 ijms-22-09722-f001:**
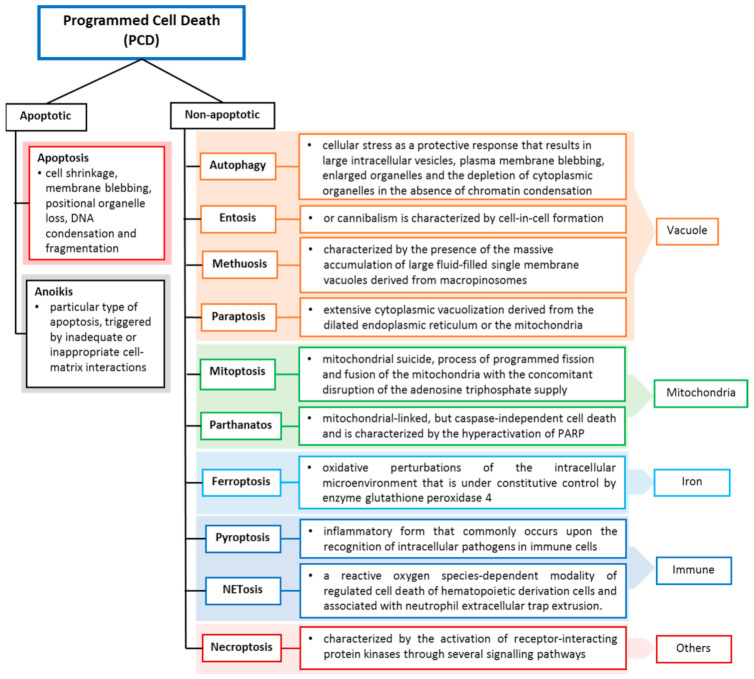
Programmed cell death classification and definitions. Cell deaths are categorized according to their morphological properties, signal-dependency and molecular mechanisms [2].

**Figure 2 ijms-22-09722-f002:**
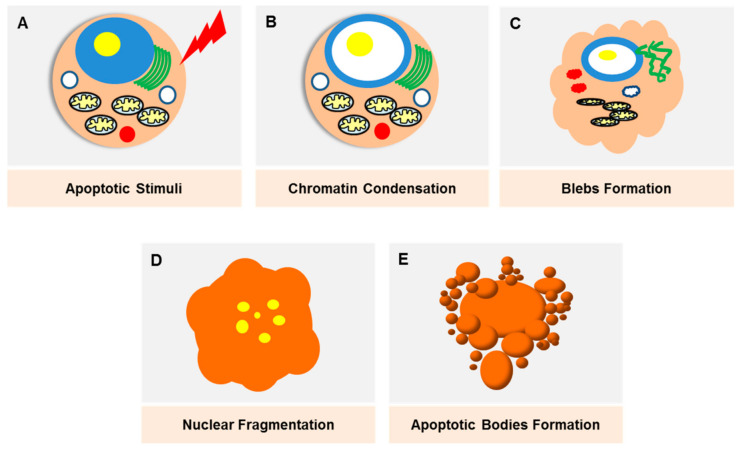
Damaged cells, stress stimuli or other triggering signals will initiate the apoptotic pathway. (**A**) The onset of apoptosis is characterised by chromatin condensation (**B**) as well as nuclear condensation. The loss of microvilli creates protrusions of the plasma membrane (blebs) (**C**). Nuclear condensation starts peripherally and finally results in fragmentation, which is also known as karyorrhexis (**D**). As the cell shrinks, the blebs separate, forming apoptotic bodies which are densely packed with cellular organelles and DNA fragments (**E**). These bodies will be rapidly phagoscytosed into macrophages and parenchymal cells without triggering an inflammatory response.

**Figure 3 ijms-22-09722-f003:**
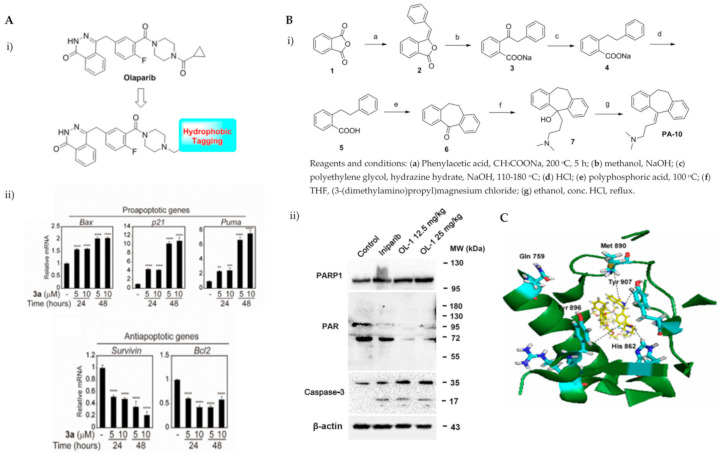
(**A**). (**i**) The strategy used to develop small molecule 3a, a PARP hydrophobic tagged-molecule. (**ii**) The expression of proapoptotic and antiapoptotic target genes measured using quantitative RT-PCR in TNBC cells treated with either vehicle or small molecule 3a at indicated times. **** *p* < 0.001. (**B**). (**i**) The general synthesis of compound PA-10 (3-(10,11-dihydro-5*H*-dibenzo[a,d] [7]annulen-5-ylidene)-*N,N*-dimethylpropan-1-amine), the lead compound used to synthesize OL-1. (**ii**) Western blot analysis of PARP, PAR and caspase-3 in tumour tissues excised from the MDA-MB-436 xenograft treated with PBS, Iniparib (100 mg/kg/d), low dose (12.5 mg/kg/d) and high dose (25 mg/kg/d) of OL-1. (**C**). Binding of conjugate 3 in the C-terminal catalytic domain of PARP1, obtained from molecular docking studies. Best pose of conjugate 3 (yellow sticks) forming hydrogen bond interactions (dotted lines) with key amino acid residues (cyan sticks) in NI, PH, and AD sites. PARP1 is shown as green ribbon. Figure 3A was reprinted with permission from Ref. [26]. Copyright (2020) European Journal of Medicinal Chemistry. Figure 3B was reprinted with permission from Ref. [27]. Copyright (2016) Springer Nature. Figure 3C was reprinted with permission from Ref. [28]. Copyright (2019) ACS publications.

**Figure 4 ijms-22-09722-f004:**
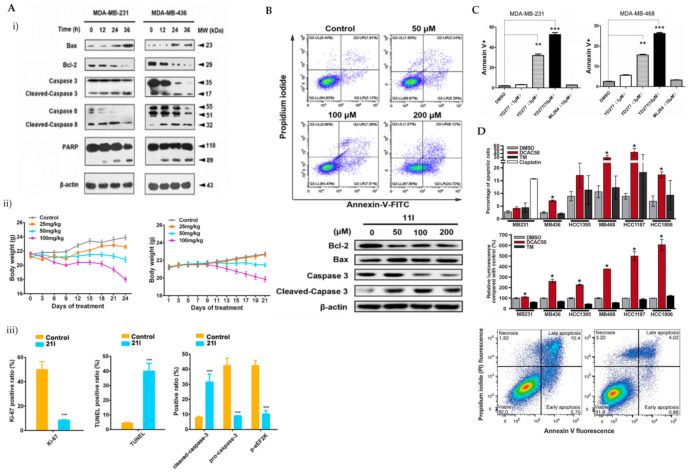
(**A**) (**i**) Western blot analysis of apoptosis-related proteins in MDA-MB-231 and MDA-MB-436 cells treated with 21l; (**ii**) tumour volumes changes observed in MDA-MB-231 or MDA-MB-436 cell xenograft mice treated with various doses of 21l; (**iii**) quantitative changes of pro- and cleaved-caspase-3 and p-eEF2K observed in the immunohistochemistry of tumour tissues excised from the control- and median-dose group treated MDA-MB-231 xenograft mice. *** *p* < 0.001 (**B**) Apoptotic cells detected using Annexin V/PI co-staining and western blot analysis of apoptotic-related proteins in MDA-MB-231 cells were treated with DMSO and compound 11l for 24 h. (**C**) Apoptotic cells detected using Annexin V/PI co-staining in MDA-MB-231 and MDA-MB-468 cells treated with various doses of YD277. DMSO serves as negative control whilst ML-264 is the parent compound. ** *p* < 0.01, *** *p* < 0.001 (**D**) Percentage of apoptotic cells detected using Annexin V/PI co-staining in TNBC cells treated with 20 mmol/L DCAC50, 30 mmol/L tetrathiomolybdate (copper chelator) or 20 mmol/L cisplatin (positive control) for 72 h. * *p* < 0.05 versus DMSO control, according to the two-tailed Student t test. Figure 4A was reprinted with permission from Ref. [38]. Copyright (2018) European Journal of Medicinal Chemistry. Figure 4B was reprinted with permission from Ref. [39]. Copyright (2020) European Journal of Medicinal Chemistry. Figure 4C was reprinted with permission from Ref. [40]. Copyright (2019) Molecular Cancer Therapeutic.

**Figure 5 ijms-22-09722-f005:**
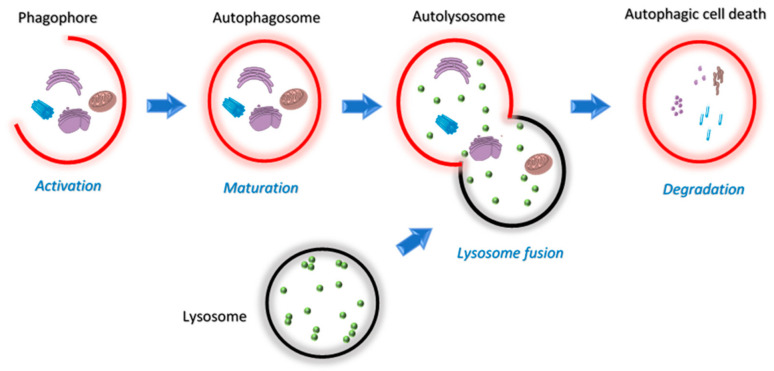
Schematic overview of the four stages of the autophagy pathway; activation, maturation, lysosome fusion and degradation.

**Figure 6 ijms-22-09722-f006:**
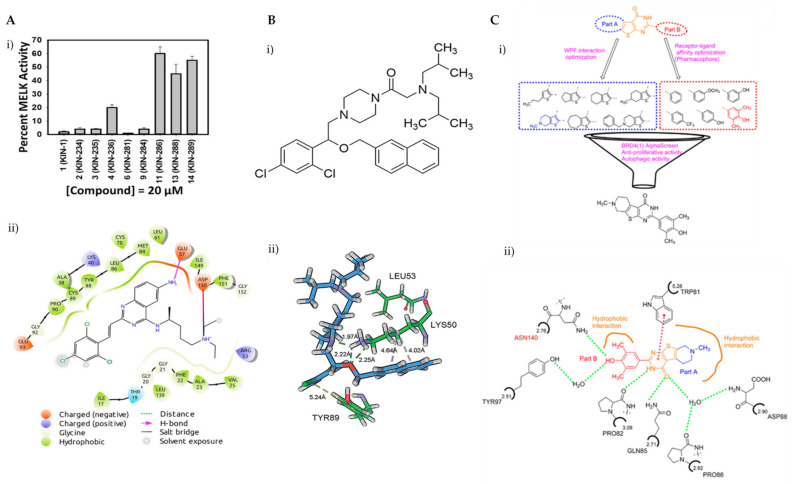
(**A**) (**i**) Several derivatives of KIN at 20 μM were tested for the inhibition of MELK, revealing KIN-281 as the best inhibitor; (**ii**) a 2D view of the interactions of KIN-281 with MELK, generated using Schrödinger Maestro. The amino acid residues are represented as droplets, with the corresponding interactions indicated in various colors, as shown in the legend. (**B**) (**i**) Chemical structure of LYN-1604; (**ii**) view of non-covalent interactions of LYN-1604 with ULK1 via hydrophobic and hydrogen bonding. Reproduced from Ref. [92] with permission from the Royal Society of Chemistry. (**C**) (**i**) Structure optimisation to identify FL-411 as a potential BRD4 inhibitor; (**ii**) non-covalent interactions of FL-411 with BRD4. Figure 6A was reprinted with permission from Ref. [86]. Copyright (2017) Bioorganic and Medicinal Chemistry. Figure 6B was reprinted with permission from Ref. [90]. Copyright (2017) The Royal Society of Chemistry. Figure 6C was reprinted with permission from Ref. [91]. Copyright (2017) American Chemical Society.

**Figure 7 ijms-22-09722-f007:**
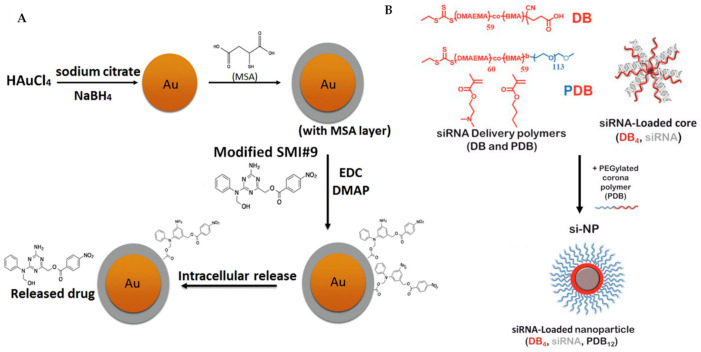
(**A**) Synthesis of SMI#9-gold nanoparticle was performed, starting from the reduction of chloroauric acid with sodium borohydride, followed by capping using mercaptosuccinic acid (MSA) and finally ester coupling of the modified SMI#9 using 1-ethyl-3-(3-dimethylaminopropyl)carbodiimide (EDC)/4-dimethylaminopyridine (DMAP). This nanoparticle undergoes endocytosis and releases the drug SMI#9 in the cell. (**B**) Three-component nanoparticles for delivery of siRNA, using core-forming polymer (poly(DMAEMA-co-BMA), DB_4_) and PEGylated and corona-forming polymer (PEG-b-poly(DMAEMA-co-BMA), PDB_12_). Figure 7A was reprinted with permission from Ref. [155]. Copyright (2016) Nanomedicine: Nanotechnology, Biology, and Medicine. Figure 7B was reprinted with permission from Ref. [156]. Copyright (2018) Cancer Research.

**Table 1 ijms-22-09722-t001:** Selected compounds targeting apoptosis and autophagy that are discussed in this review.

Apoptosis			
**Structure**	**Code/Name**	**Pathway Affected**	**Refs**
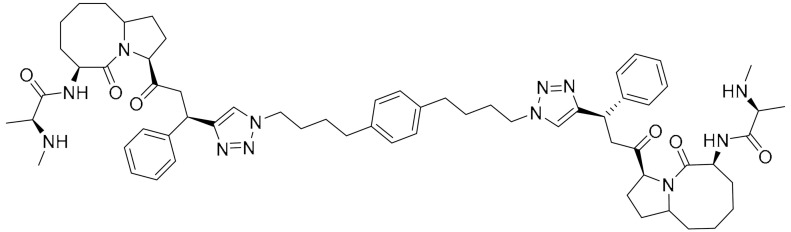	SM-164	mediate caspase activation via inhibition of XIAP and cIAP-1	[18]
nanodelivery	nanoliposomal (NL)-Bcl-2 siRNA	reduce the expression of Bcl-2	[20]
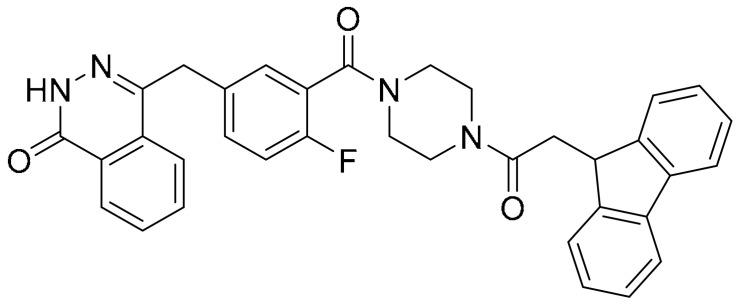	Compound 3a	induce PARP1 degradation	[26]
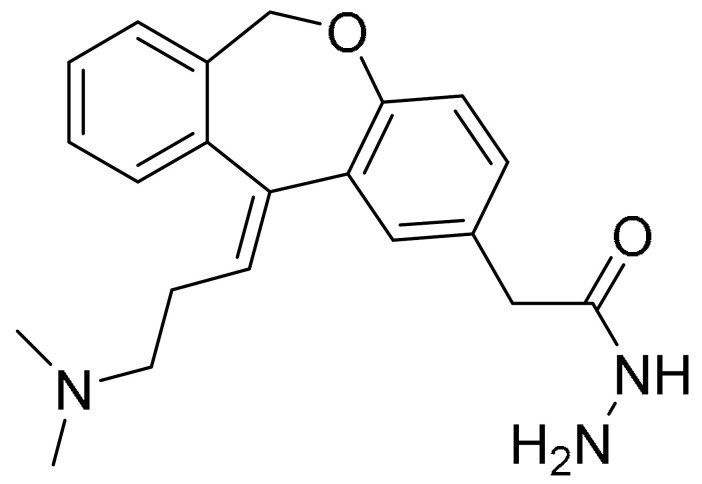	OL-1	PARP1 inhibition	[27]
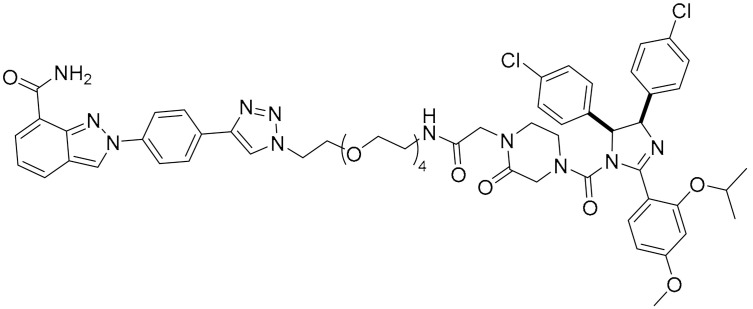	Compound 3	induce proteasome-dependent PARP1 cleavage	[28]
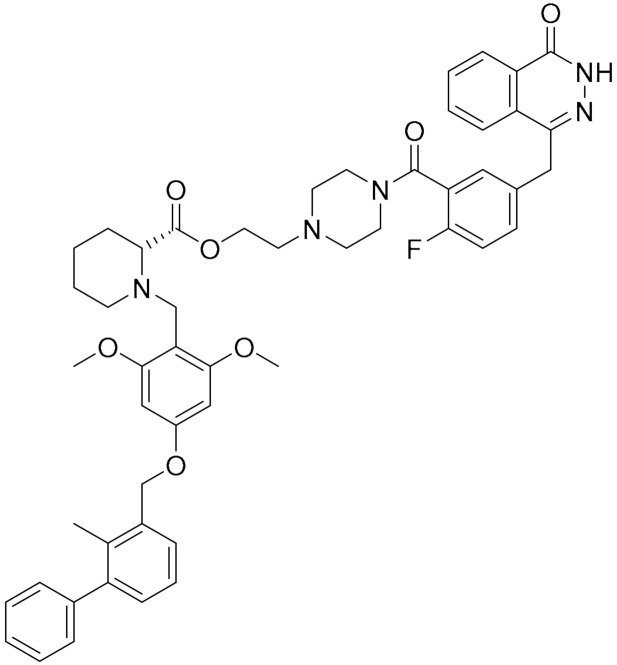	Conjugate 3	target PARP1 and PD-L1	[29]
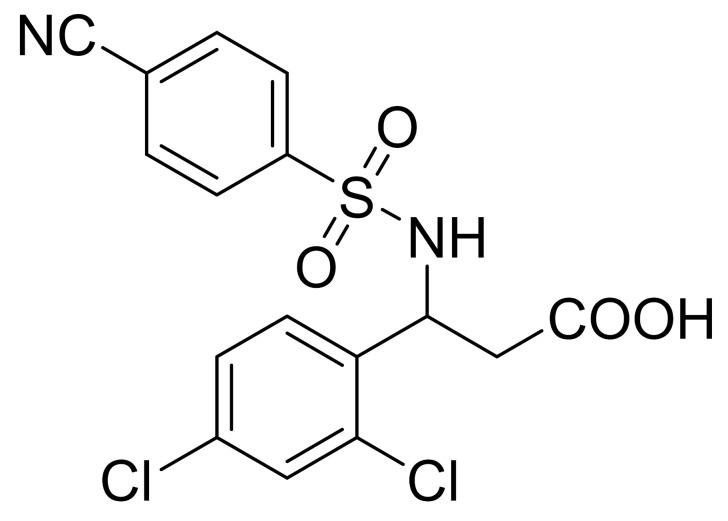	21l	eEF2K inhibitor	[38]
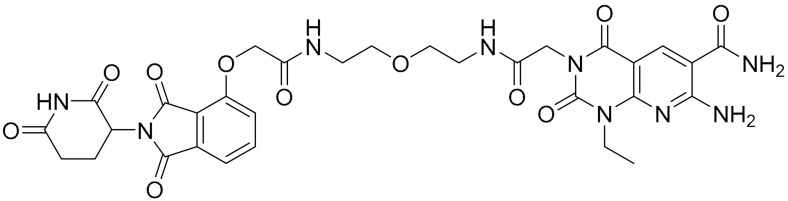	11l	degradation via ubiquitin-proteasome-Mediated eEF2K and p-eEF2K	[39]
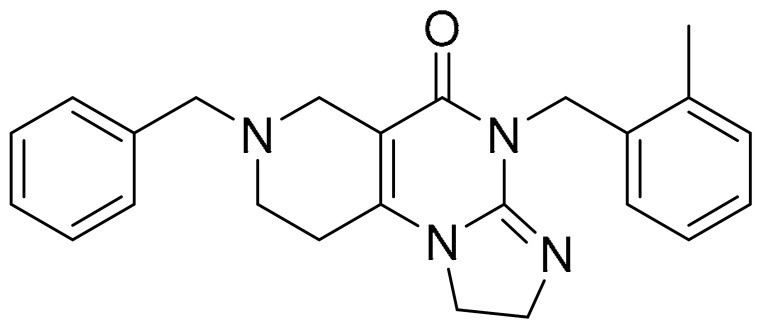	ONC201	activate ATF4/CHOP-mediated ER stress response	[46]
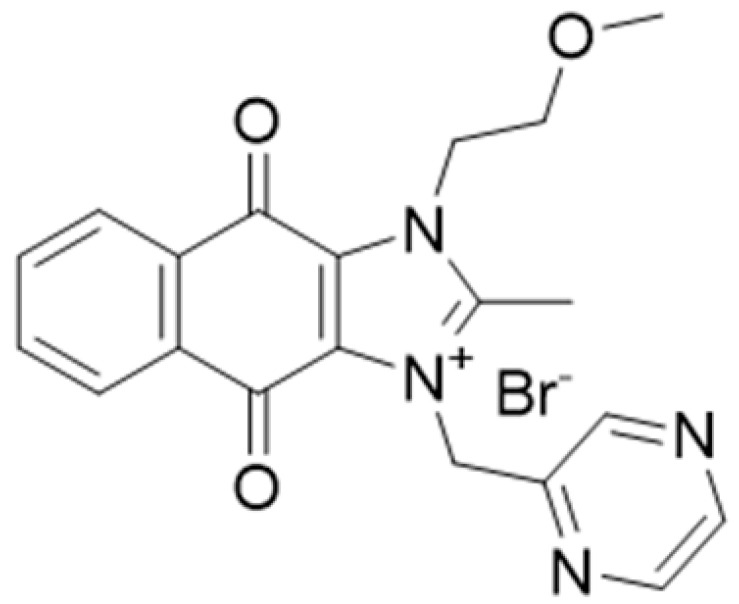	YM155	upregulates DR5 expression	[48]
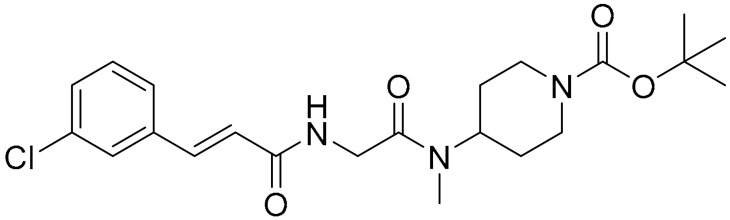	YD277	induce IRE1α transcription which leads to JNK activation	[50]
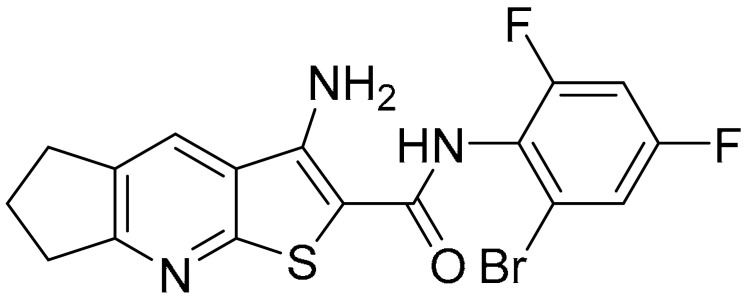	DCAC50	ATOX1 and CCS inhibitor	[40]
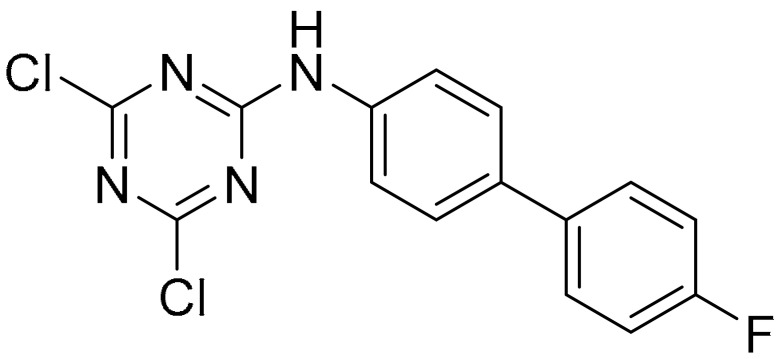	KEAI-97	interaction disruption between thioredoxin and caspase-3	[54]
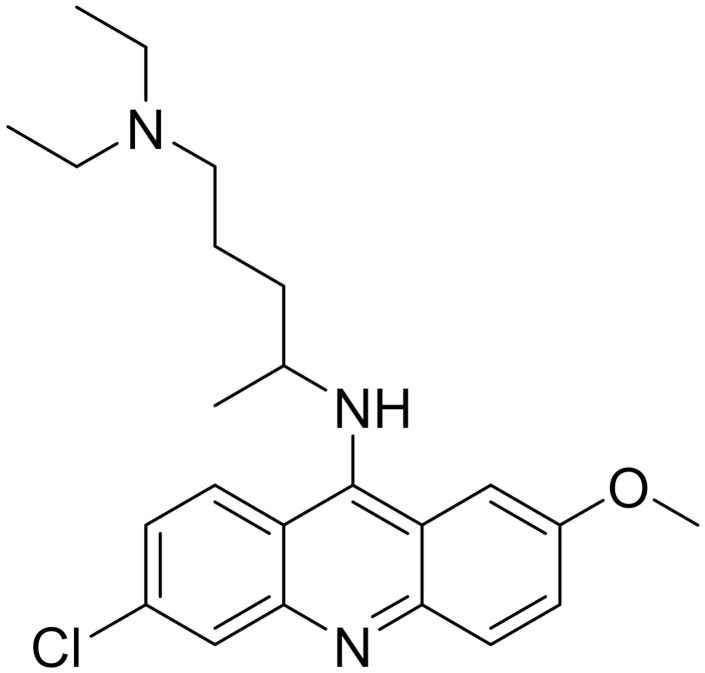	Quinacrine	S-phase arrest and upregulates p53	[56]
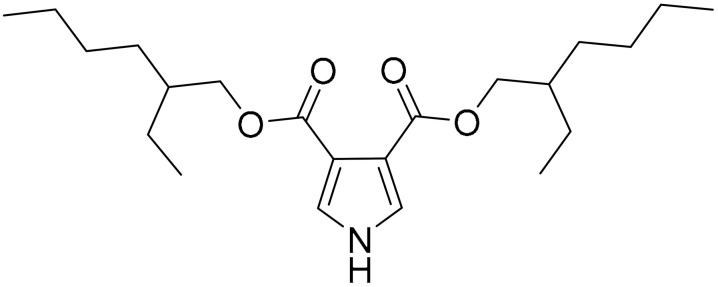	bis(2-ethylhexyl)-1H-pyrrole-3,4-dicarboxylate (TCCP)	elevates ROS and intracellular Ca^2+^ ion concentration	[58]
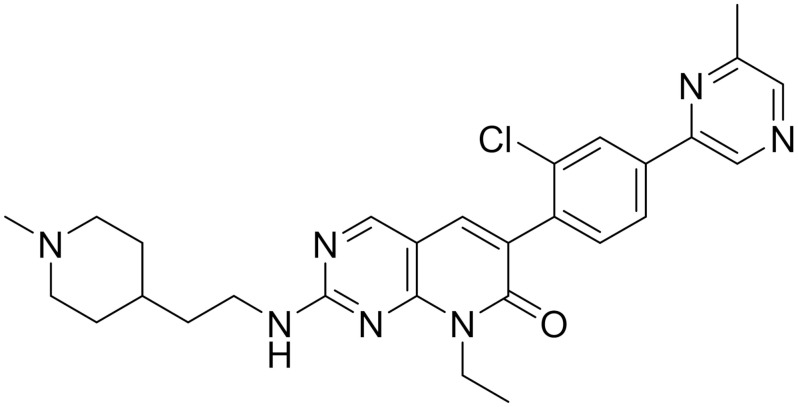	FRAX1036	increases PARP cleavage and reduces cyclin D1 expression	[59]
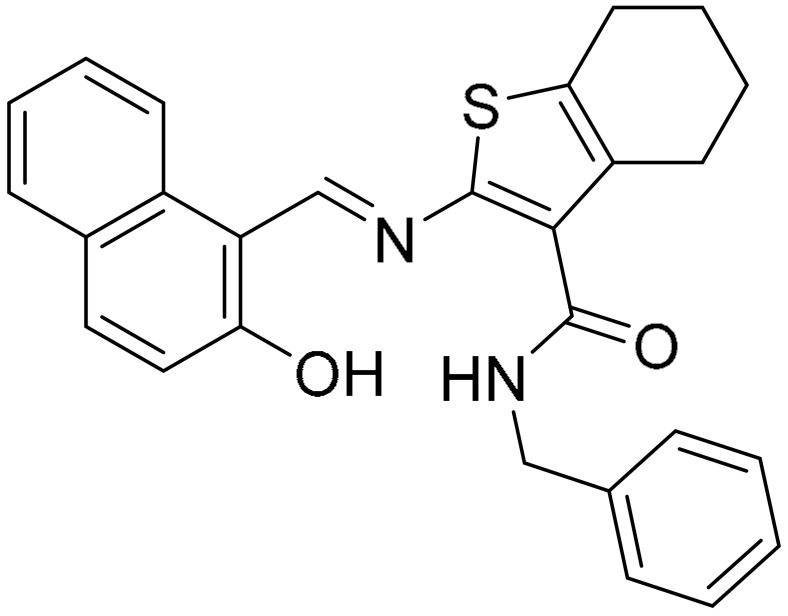	JGB1741	SIRT1 inhibitor	[61]
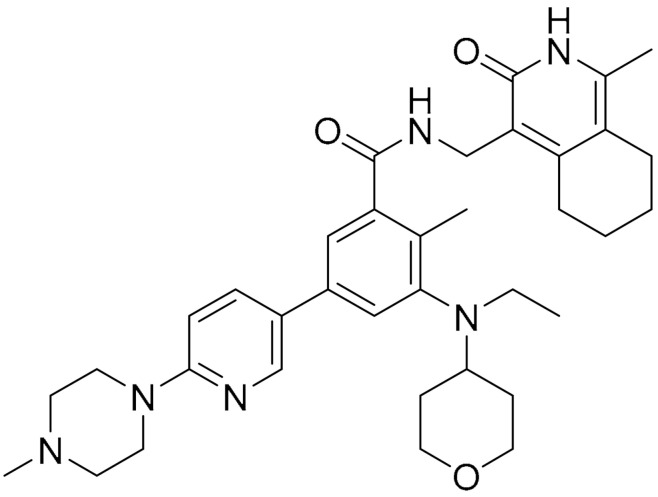	ZLD1039	EZH2 inhibitor	[63]
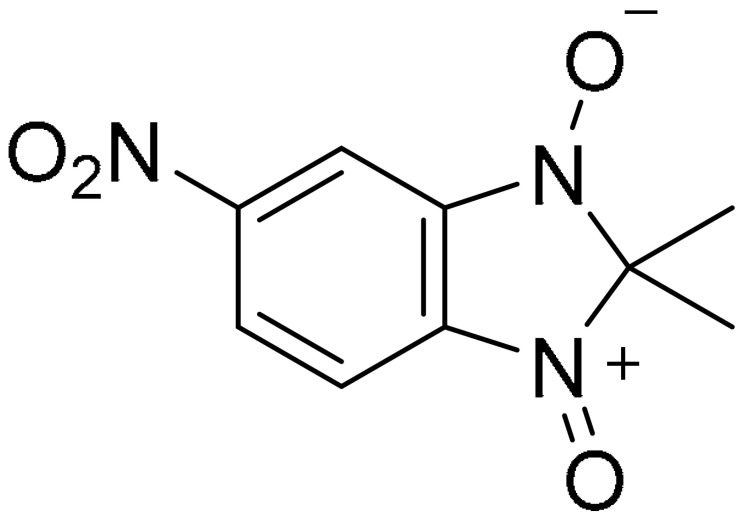	Sepin-1	separase inhibitor	[65]
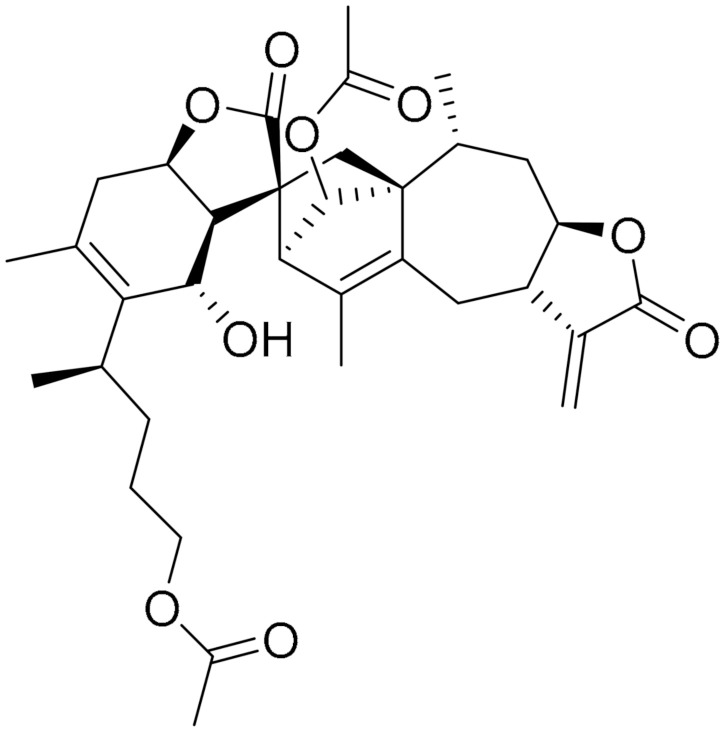	Inulanolide A (InuA)	cell cycle arrest at G2/M phase	[67]
**Autophagy**			
Structure	Code/name	Target	Refs
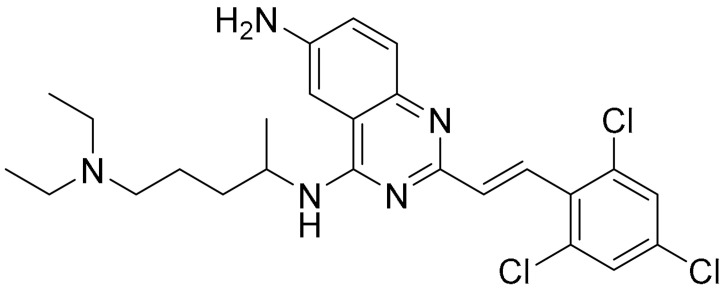	KIN-281	multiple kinases (maternal leucine zipper kinase (MELK) and the non-receptor tyrosine kinase bone marrow X-linked (BMX))	[86]
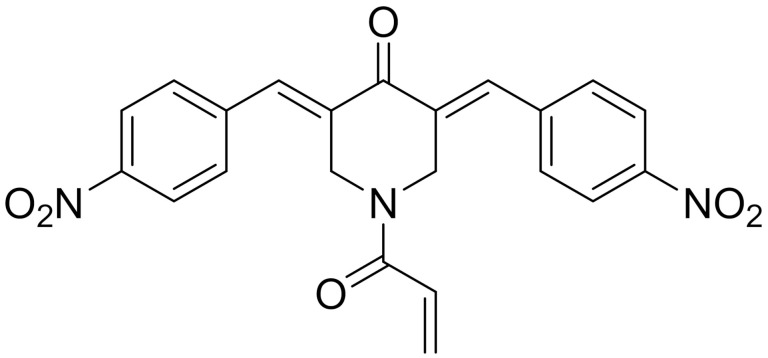	b-AP15	deubiquitinating enzymes	[87]
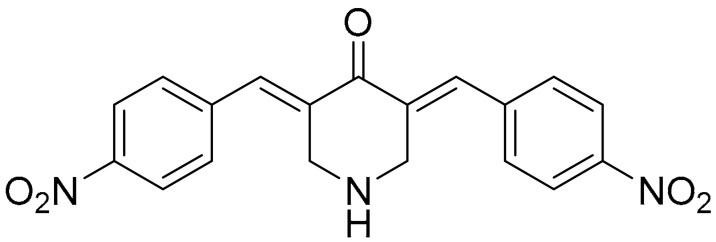	RA-9		
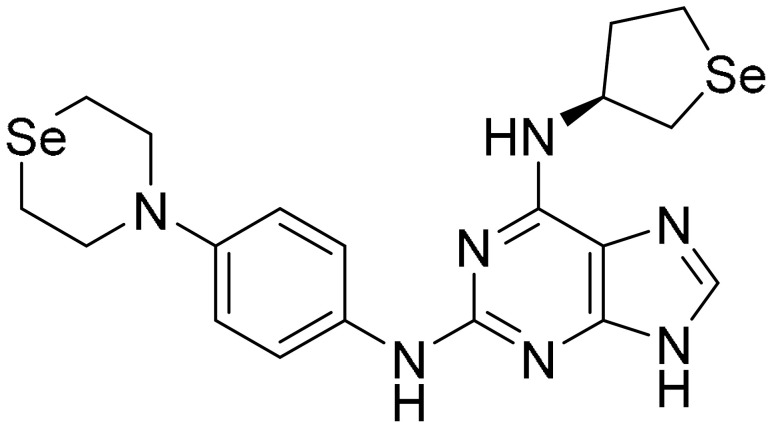	SLLN-15	AKT-mTOR	[88]
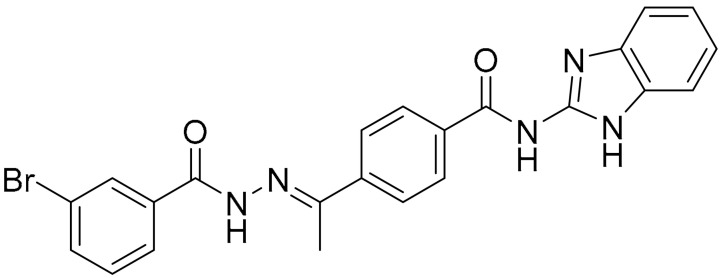	(*E*)-N-(1H-benzo[d]imidazol-2-yl)-4-(1-(2-(3-bromobenzoyl)hydrazineylidene)ethyl)benzamide	mTOR	[89]
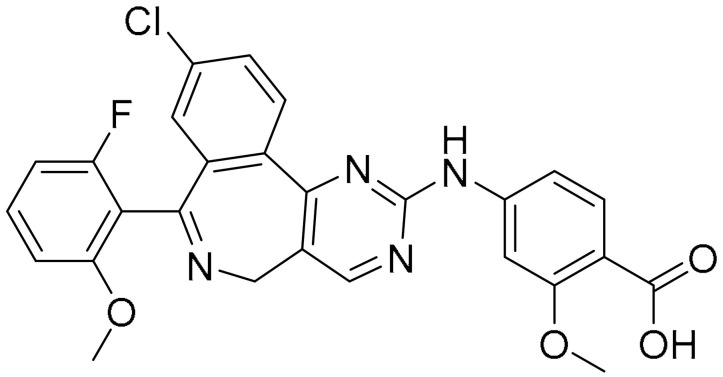	MLN8237	Aurora A (AURKA) kinase	[92]
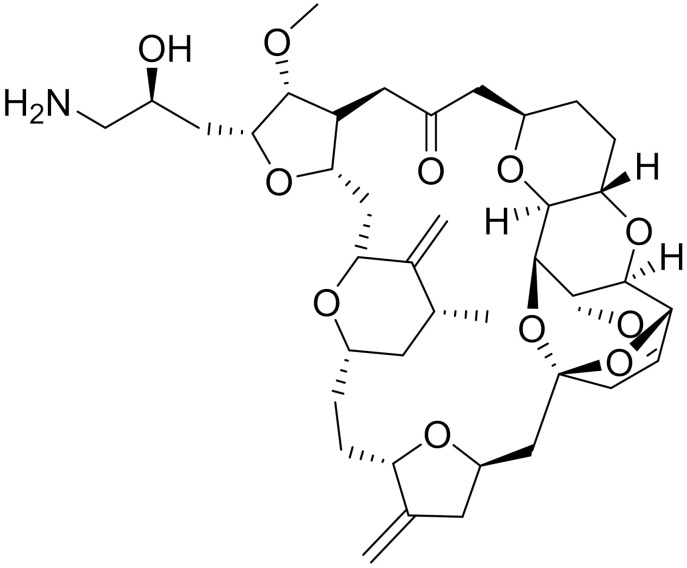	Eribulin		
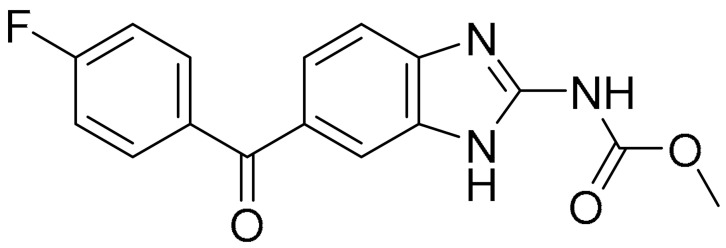	Flubendazole	Protein 4B(ATG4B)	[93]
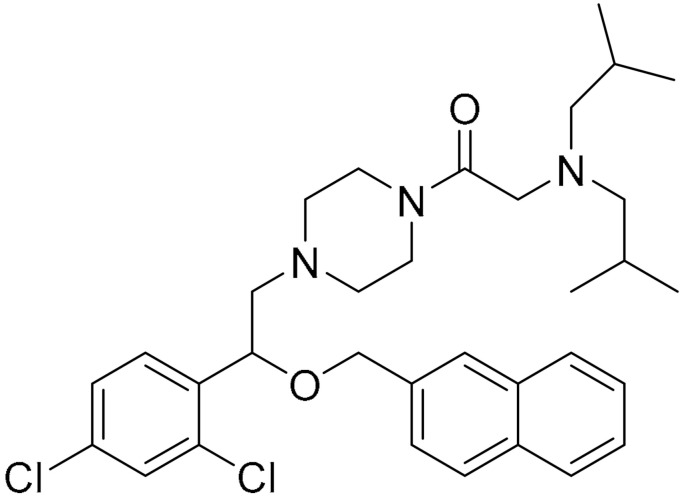	LYN-1604	UNC-51-like kinase 1 (ULK1)	[90]
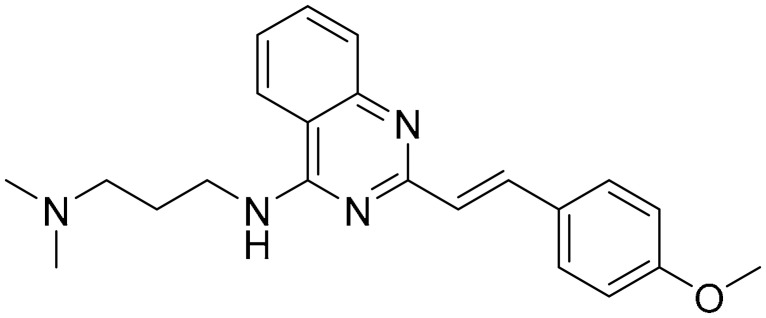	CP-31398	p53	[94,95,96]
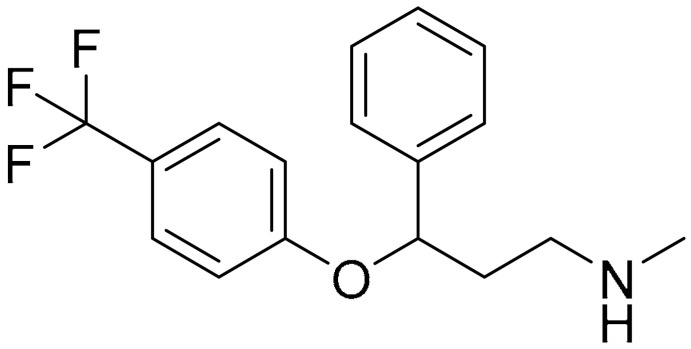	Fluoxetine	eEF2K	[97]
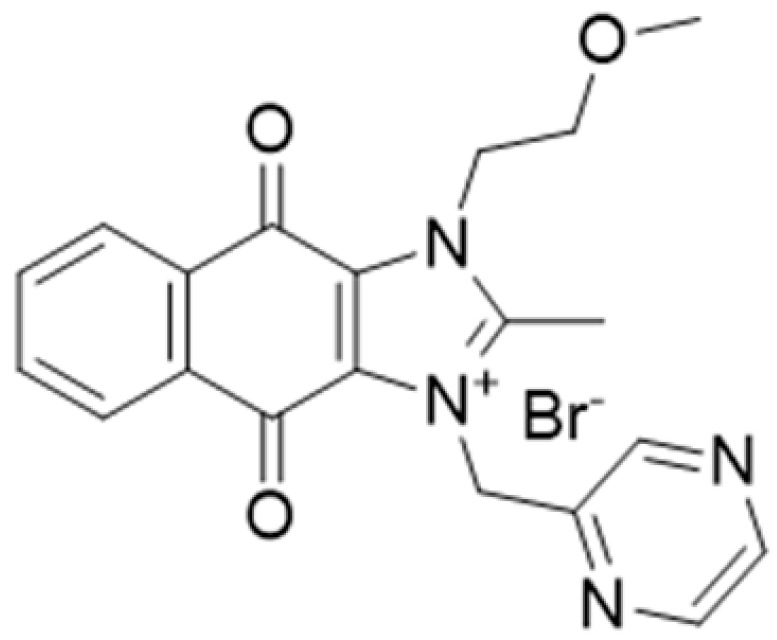	YM155	Survivin	[102]
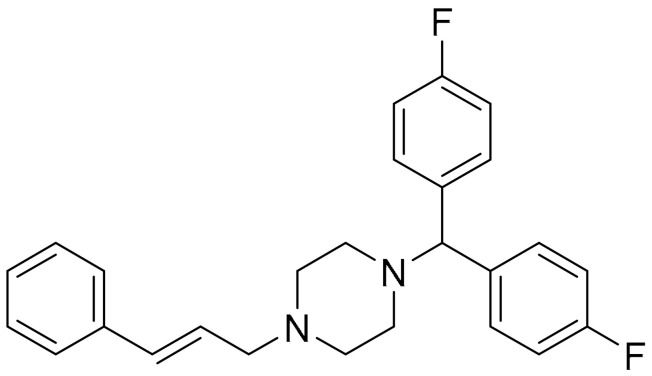	FLN	N-Ras	[106]
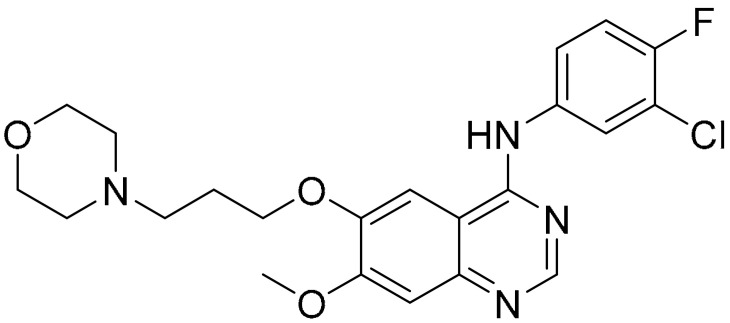	Gefitinib (Iressa^®^, ZD1839)	epidermal growth factor receptor (EGFR) tyrosine kinase	[110]
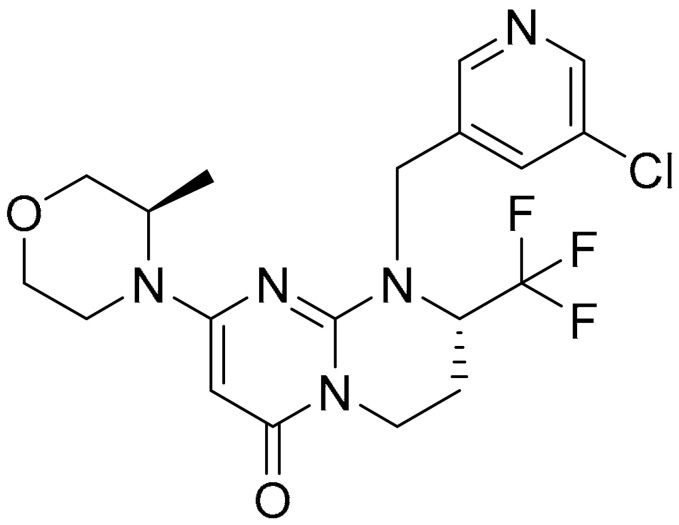	SAR405	Vps34 p110α and/or HER2	[112]
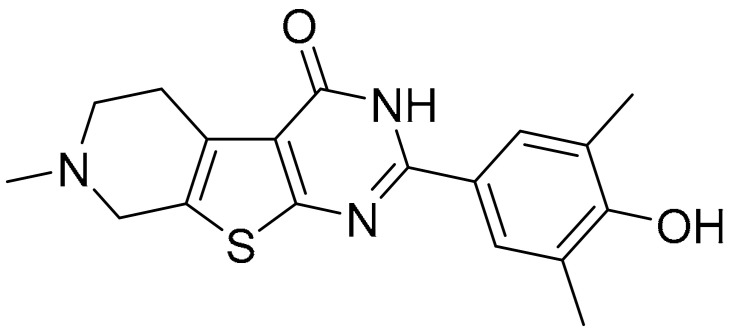	FL-411	Bromodomain-containing protein 4	[91]
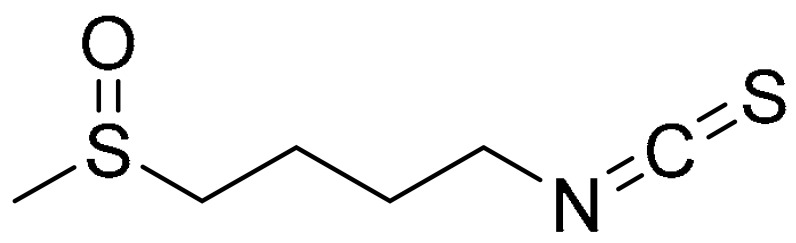	Sulforaphane	Histone deacetylase (HDAC)	[124,125]
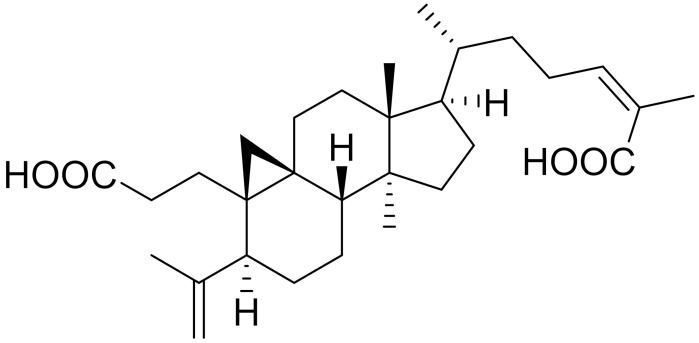	PC3-15	Ubiquitin enzymes	[127,128]
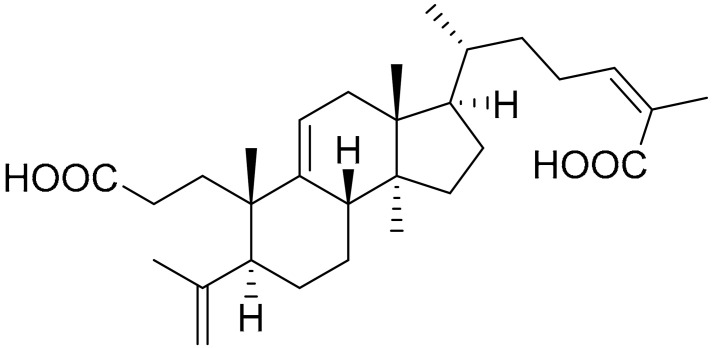	PC3-16		
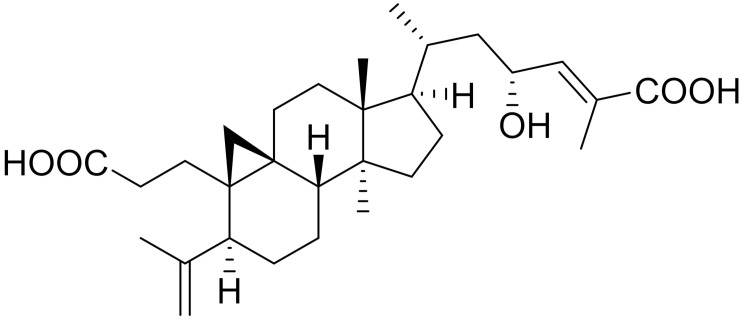	PC3-17		
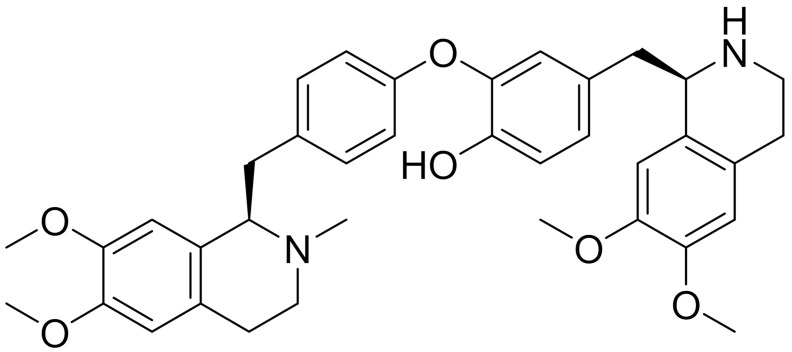	*N*-Desmethyldauricine (LP-4)	SERCA	[129]
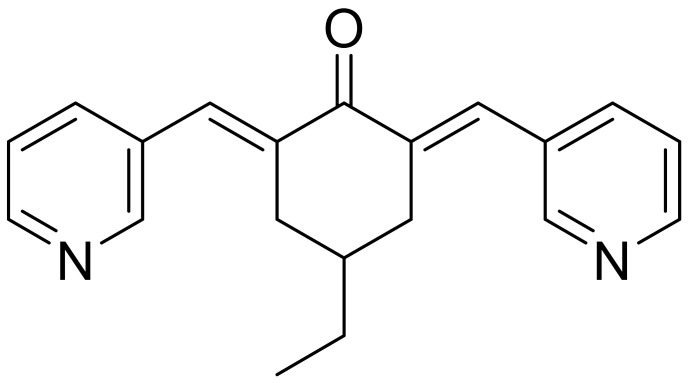	MCB-613	steroid receptor coactivators	[132]
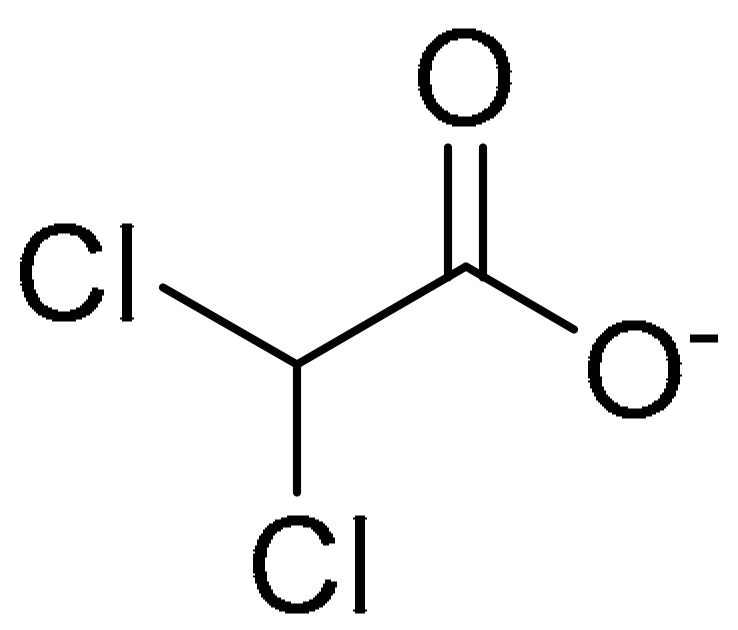	Dichloroacetate	enzyme pyruvate dehydrogenase kinase	[137]
structure not known	SB02024	Vps34	[142]
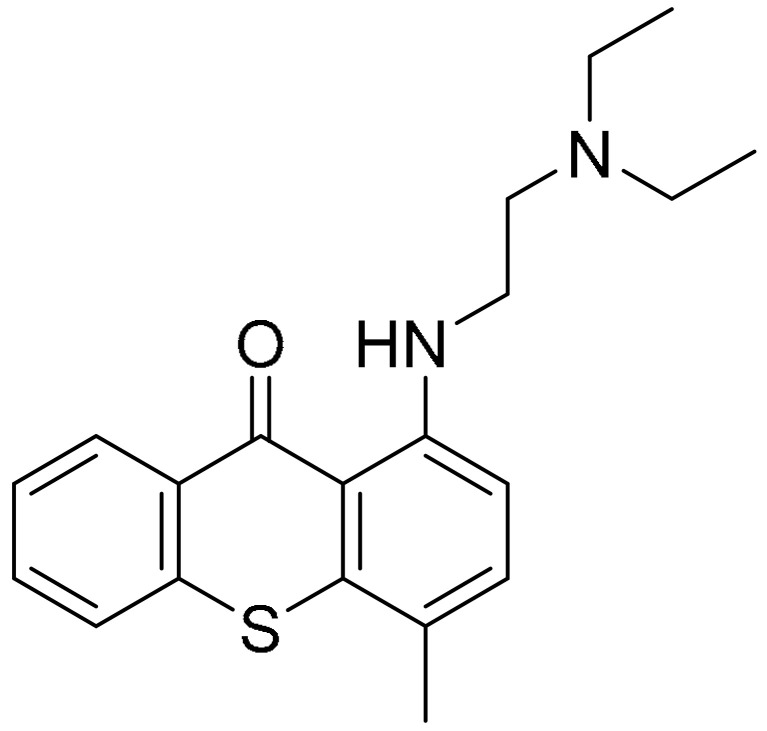	Lucanthone (Miracil D)	Cathepsin-D	[145]
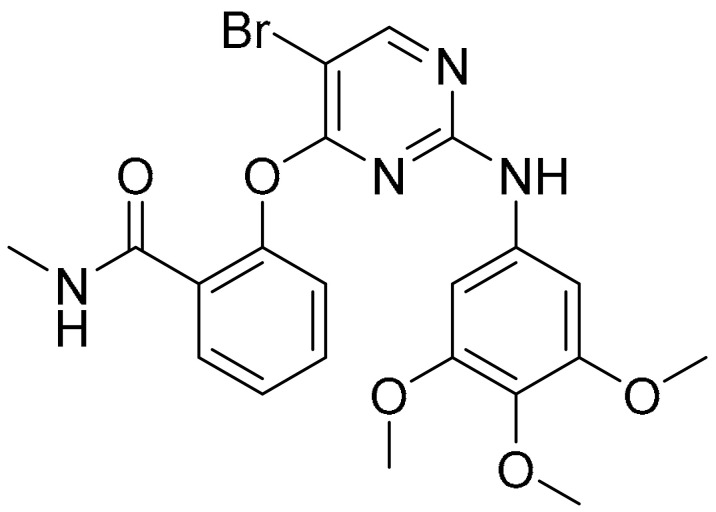	SBI-0206965	ULK1	[146]
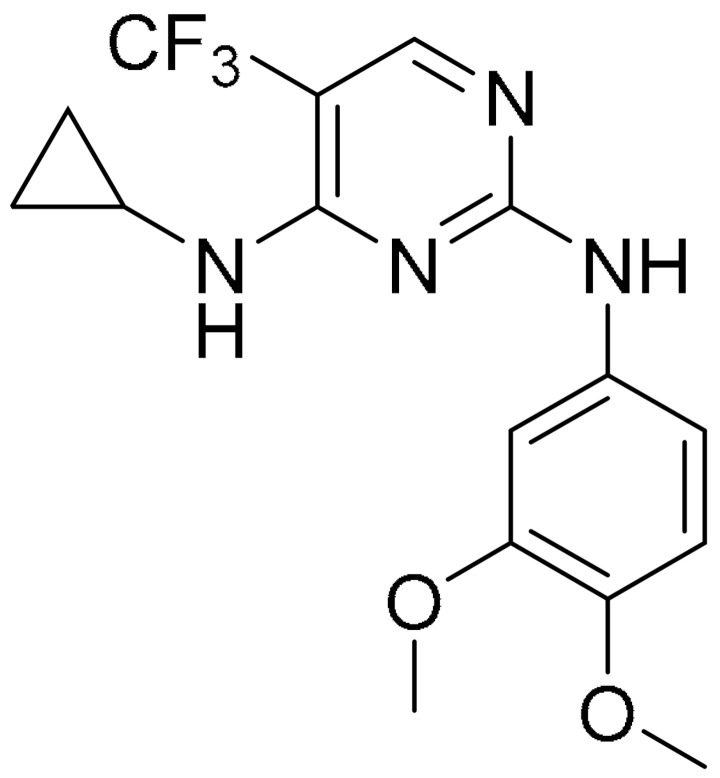	SBP-7455	ULK1/2 and inhibits their enzymatic activity	[147]
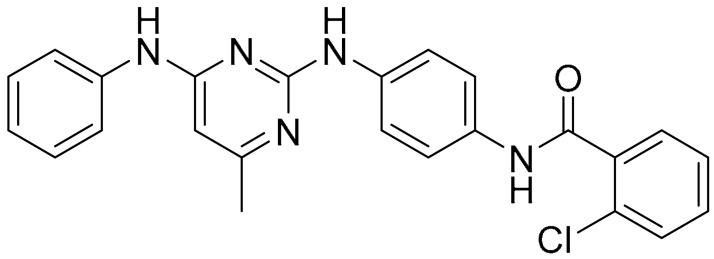	Aumitin	mitochondrial respiration complex 1	[148]
Nanoparticle	SMI#9-gold nanoparticles	Rad6	[155]
nanoparticle	Composed of siRNA, (poly(DMAEMA-co-BMA), DB4), and PEGylated, (PEG-b-poly(DMAEMA-co-BMA),	kinase mTORC2	[156]
nanoparticle	Poly (β-amino ester) and poly (ethylene glycol) (PEG)	Beclin-1	[167]

## Data Availability

Not applicable.

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
