# Peer review of "Small Molecules Targeting Programmed Cell Death in Breast Cancer Cells"

_ijms, 2021, doi:10.3390/ijms22189722_

Round 1

Reviewer 1 Report

This review discusses the mechanisms of programmed cell death and small molecule treatment in breast cancer.
First the authors describe the subtype of programmed cell death. Next the authors presented results of their experiments focusing on the apoptosis by small molecule treatment. The review part is organized well, but there is too much their experimental data in the manuscript. These results should be reviewed by peer-reviewer.
The authors must separate the review part and their experimental results, and I strongly recommend to submit their results as an article.
After acceptance of the article, the authors should submit the review with citing the article.
I do not recommend this review for the publication in the International Journal of Molecular Sciences.

Author Response

Reviewer 1

This review discusses the mechanisms of programmed cell death and small molecule treatment in breast cancer.

First the authors describe the subtype of programmed cell death. Next the authors presented results of their experiments focusing on the apoptosis by small molecule treatment.

Yes, we agree that the first part is a description of programmed cell death. However, the reviewer has completely mistaken to the subsequent part as being our experimental result. The next section is in fact a critical review of apoptosis in TNBC. We have clearly outlined the review at the end of the introduction section:

“This review will highlight the small molecules with promising preclinical data that target autophagy and apoptosis to induce cell death in TNBC cells. Here, we begin with the discussion of the recent preclinical research and developments of small molecules that affect the apoptotic, genomic instability, proliferating and stress-mediated signalling pathways of TNBC cells that lead to apoptosis. Next, we focus on small molecules that inhibit and enhance autophagy pathways in TNBC and other breast cancer cells through selectivity and sensitivity, and enhanced delivery methods. Finally, we summarise the perspective and potential of small molecules in promoting cell death through apoptosis and autophagy.”

The review part is organized well, but there is too much their experimental data in the manuscript. These results should be reviewed by peer-reviewer.

We thank the reviewer for acknowledging that the review is well organised. With respect, the reviewer has mistaken in identifying the discussion of our section as our experimental results. They are indeed not. We have clearly indicated in each section that we are highlighting the present work in the field, as seen in the paragraphs below.

End of section 2

“….In this review, small molecules that target pivotal signalling pathways involved in modulating hallmarks of cancer attributes which include evading apoptosis, genomic instability, sustaining proliferative signalling and alteration in the tumour microenvironment will be highlighted. In particular, emphasis is placed on data obtained in TNBC research.”

End of section 3

“…. Here, we would like to highlight some of the small molecules that are selective and sensitive towards inducing and inhibiting autophagy TNBC and other breast cancer cells.”

The authors must separate the review part and their experimental results, and I strongly recommend to submit their results as an article.

Again, with respect, the reviewer has mistaken the discussion to be our experimental results which they are clearly not. Each of the results presented has been referenced accordingly.

After acceptance of the article, the authors should submit the review with citing the article. I do not recommend this review for the publication in the International Journal of Molecular Sciences.

The reviewer has yet again mistaken to suggest that the results presented in the review should be submitted to a peer-reviewed article.  

Reviewer 2 Report

This is a review regarding small molecules that target apoptosis and autophagy with promising preclinical data by inducing cell death in breast cancer cells, in particular TNBC cells.  

This comprehensive review is very well written and covers more than 40 small molecules able to induce cell death in breast cancer cells. The authors cover 173 references, several of them from the past few years.

Minor comments and suggestions:

  • It would be helpful to include a Table in the beginning of the article with all the small molecules and their targets that will be presented in the review.
  • Figure 3B, i)- the legend under the figure is difficult to read.
  • Lines 228-239- paragraph “Two different strategies...positive control”: this paragraph is not clear enough. Please, clarify what BMS-001 is and add references to the paragraph.
  • Line 328: Define ER when it is used for the first time.
  • Figure 4: The letter “D” is missing in the figure.
  • Figures 4 C second and third graphics: it is difficult to read the name of the cell lines under the columns.

Author Response

Reviewer 2

This is a review regarding small molecules that target apoptosis and autophagy with promising preclinical data by inducing cell death in breast cancer cells, in particular TNBC cells.  

This comprehensive review is very well written and covers more than 40 small molecules able to induce cell death in breast cancer cells. The authors cover 173 references, several of them from the past few years.

We thank the reviewer for acknowledging that the review is well written and comprehensive covering recent advancement in the field.

Minor comments and suggestions:

  • It would be helpful to include a Table in the beginning of the article with all the small molecules and their targets that will be presented in the review.

We thank the reviewer for this suggestion, and we have included a table representing all the compounds and their targets (Table 1).

  • Figure 3B, i)- the legend under the figure is difficult to read.

The legend for Figure 3B has been corrected.

  • Lines 228-239- paragraph “Two different strategies...positive control”: this paragraph is not clear enough. Please, clarify what BMS-001 is and add references to the paragraph.

The paragraph has been re-written to clarify the points.

Line 328: Define ER when it is used for the first time.

The word endoplasmic reticular (ER) has been abbreviated in line 182.

  • Figure 4: The letter “D” is missing in the figure.

The letter ‘D’ has been added.

  • Figures 4 C second and third graphics: it is difficult to read the name of the cell lines under the columns.

These have been corrected accordingly.

Round 2

Reviewer 1 Report

First, I apologize the my misreading of the manuscript.
I understood that it is not the results by the authors.
If so, the authors will have the following problems: The author's use a lot of the data from cited article, isn't it infringing on the copyright?
I have never seen a review that uses so much cited data in the biology and medicine fields.
In addition, the authors should obtain permission to reprint in advance.

I recommend to use original animation or figures which summarize the results of cited article.

Author Response

Thank you very much for your comment. We believe that the biological data cited in this manuscript is important to highlight the findings of the small molecule. We also identified few other review articles published in reputable journals which use similar practice.

Nanotechnology for Multimodal Synergistic Cancer Therapy. Wenpei Fan, Bryant Yung, Peng Huang, and Xiaoyuan Chen. Chemical Reviews 2017 117 (22), 13566-13638. DOI: 10.1021/acs.chemrev.7b00258

Magnetic Resonance Spectroscopy in Metabolic and Molecular Imaging and Diagnosis of Cancer. Kristine Glunde, Dmitri Artemov, Marie-France Penet, Michael A. Jacobs, and Zaver M. Bhujwalla. Chemical Reviews 2010 110 (5), 3043-3059. DOI: 10.1021/cr9004007

We have obtained the permissions from the respective publishers to use a part of their figure. We have attached these documents during the submission process. We do understand that this approval from the publisher is not clearly stated in the manuscript. Hence, we have added a copyright statement for all the relevant figures.